# Malaria trends in Ethiopian highlands track the 2000 'slowdown' in global warming

Xavier Rodó [1,7], Pamela P. Martinez [2,6], Amir Siraj[3] & Mercedes Pascual [4,5,7 ✉]

A counterargument to the importance of climate change for malaria transmission has been that regions where an effect of warmer temperatures is expected, have experienced a marked decrease in seasonal epidemic size since the turn of the new century. This decline has been observed in the densely populated highlands of East Africa at the center of the earlier debate on causes of the pronounced increase in epidemic size from the 1970s to the 1990s. The turnaround of the incidence trend around 2000 is documented here with an extensive temporal record for malaria cases for both *Plasmodium falciparum* and *Plasmodium vivax* in an Ethiopian highland. With statistical analyses and a process-based transmission model, we show that this decline was driven by the transient slowdown in global warming and associated changes in climate variability, especially ENSO. Decadal changes in temperature and concurrent climate variability facilitated rather than opposed the effect of interventions.

[1] ICREA and CLIMA (Climate and Health) Program, ISGlobal, Barcelona, Spain. [2] Department of Epidemiology, Center for Communicable Disease Dynamics, T.H. Chan School of Public Health, Harvard University, Boston, MA, USA. [3] Department of Biological Sciences, University of Notre Dame, Notre Dame, IN, USA. [4] Department of Ecology and Evolution, University of Chicago, Chicago, IL, USA. [5] Santa Fe Institute, Santa Fe, NM, USA. [6] Present address: Department of Microbiology and Department of Statistics, University of Illinois at Urbana, Champaign, Champaign, IL, USA. [7] These authors contributed equally: Xavier Rodó, Mercedes Pascual. ✉email: pascualmm@uchicago.edu

Global warming is expected to promote malaria transmission in highlands because temperature decreases with elevation, limiting both the abundance of mosquitoes and the development of the parasite within these vectors[1–7]. At the edge of the geographic distribution of the disease, highlands are characterized by seasonal and intermittent large epidemics, and are described as regions of "unstable" malaria transmission, where both average transmission intensity and population immunity are typically low. It is in these regions that effects of climate conditions, control efforts, and other external drivers such as immigration from lower endemic areas, should be most apparent in the temporal patterns of incidence[8]. Despite evidence that warmer temperatures exacerbated unstable malaria in densely populated highlands of East Africa from the 1980s to the 1990s[2,7], the declining trends in incidence of the following decade challenge the importance of climate forcing vs. control[4,5]. As for the long debate concerning increasing malaria incidence[1–7], effects of temperature cannot be simply extrapolated based on its known effects on parameters of vectors and parasites in the laboratory[9–11]. Multiple drivers other than climate ones might be at play, and quantification of the integrated effects of temperature is required. The turnaround from an increasing to a decreasing trend in malaria incidence at the turn of the new century provides an opportunity to further address the importance of climate forcing based on epidemiological patterns. With a longer temporal record from disease surveillance at hand, we can now ask whether changes in climate preceded and facilitated the results of extensive public health intervention.

Importantly, the turnaround in malaria incidence at the beginning of the century could reflect a strong coupling to temperatures rather than a consequence of control efforts per se. Specifically, the transient "slowdown" in the warming trend itself could have played an important role in driving reduced epidemics in the background of existing control efforts and before their enhancement. At a global scale, this slowdown was initially termed the "global warming hiatus" on the basis of a presumed lack of an increase in Global Mean Surface Temperature (GMST) over the period from 1998 to 2005 (in the HadCRUT3 data set[12,13]). It was itself the subject of controversy and debate given its wide implications[14–28]. A discrepancy between observations and climate model projections initially suggested a potential overestimation of climate sensitivity to anthropogenic forcing[16,17]. Main areas of discussion later involved the roles of the Pacific Ocean internal variability (in particular effects of the El Niño Southern Oscillation, ENSO, and the phase of the Pacific Decadal Oscillation, PDO)[18], the land masses of North America and Eurasia dominating land temperature trends[29], volcanic activity[19], and the excess of heat stored in deeper ocean layers of the Pacific Ocean, and to a minor degree, of the Atlantic Ocean[21–23]. Consensus was finally reached with the inter-calibration of monitoring sources and models[30].

The warming slowdown was largely explained as a redistribution of heat within the Earth's system[23,31], with enhanced uptake of heat energy by the global ocean involving the strongest El Niño on record for 1998 followed also by a strong La Niña. The decadal and transient pattern of variation in the rate of increase of GMST must be seen as the result of averaging at the global scale across different regions, with dynamical contributions of the major oceanic drivers of climate variability, including ENSO and the PDO[14,18,20,26,32]. A recent review conveys the importance of decadal climate variation of changes in the tropical Eastern Pacific, where ENSO originates, and the increasing evidence for complex inter-basin feedbacks between the tropical Atlantic, Indian and Pacific oceans[33]. Irrespective of the final causes behind it, the transient global warming slowdown took place at the turn of the century (1998–2012) coincident with

documented decreases in seasonal malaria epidemics in East Africa[5,7].

We, therefore, address here the link between the reversal in malaria's decadal trend in Ethiopian highlands and this concomitant slowdown in global warming by considering its regional manifestation in local temperatures. We further examine the role of main modulators of this regional climate at interannual and decadal scales, namely ENSO and the PDO respectively. We take advantage of extensive retrospective records on malaria cases from 1968 to 2007 for both parasites *Plasmodium falciparum* ($P_f$) and *Plasmodium vivax* ($P_v$) with different epidemiology in a highland region of Oromia, Ethiopia, where widespread public health interventions against malaria did not start until 2004–2005[34,35]. We also analyze the full ensemble of weather stations for this region. In addition to variance and trend decompositions of local malaria, rainfall, and temperature time series, we examine connections between malaria cases, regional climate, and global climate variability. We use an atmospheric model to prescribe the two major modes of variability, ENSO and PDO, to investigate their differential effects over the region before and after the turn of the century. These connections help establish a consistent chain of effects and assess the consistency of mechanisms and their associated variability across scales. In particular, a process-based transmission model for *P. falciparum* is used to predict what would have been the effect on malaria cases of the observed change in temperatures in the absence of the public health interventions introduced post-2004. Evidence that malaria dynamics in highland regions did indeed closely track the slowdown in temperatures and its associated global drivers, would indicate strong coupling between the disease and climate. Such coupling would have preceded and therefore facilitated public health efforts, acting synergistically to reduce malaria risk. It would have continued to act even after these efforts decreased malaria risk. Implications for the considerable extents of the East African highlands that did not experience such interventions on malaria transmission and for future reversals of current trends should be carefully considered.

## Results
The Debre Zeit study area within the Oromia region borders the East African Rift Valley and includes parts of the Central Highlands of Ethiopia. Its altitude ranges from 1600 to 2500 meters above sea level from the South East to the North West respectively (Fig. 1A). The temporal evolution of minimum temperature ($T_{min}$) is shown in Fig. 1B for the interval 1968–2007 together with its long-term mean ("Methods" section). For the regional climate data, we extended the previous composited time series[7] by averaging eleven instead of the four meteorological stations closest to the study site in Debre Zeit, to obtain monthly minimum ($T_{min}$) and maximum ($T_{max}$) temperatures, and precipitation ("Methods" section, Fig. 1B, C and Supplementary Figs. 1, 2). We refer hereafter to this new composite regional time series as $DZ_{reg}$, and to the earlier one as DZ. For comparison, for the interval from 1993 onwards, average time series for the climate variables were also generated from the entire set of 24 local climate ground stations in Oromia (hereafter ORO, Supplementary Fig. 2A). Both resulting regional time series, from the 11 and 24 stations respectively, co-evolved in high synchrony with the DZ $T_{min}$ and $T_{max}$ records (Supplementary Figs. 1 and 2). Differences appear minor and are localized with the timing of the movement up in elevation of one of the DZ stations[7] ("Methods" section). The area is epidemic-prone and characterized by unstable malaria transmission with marked variation in the number of cases among years. The main transmission season follows the (June–August) long rains and takes place from the beginning of

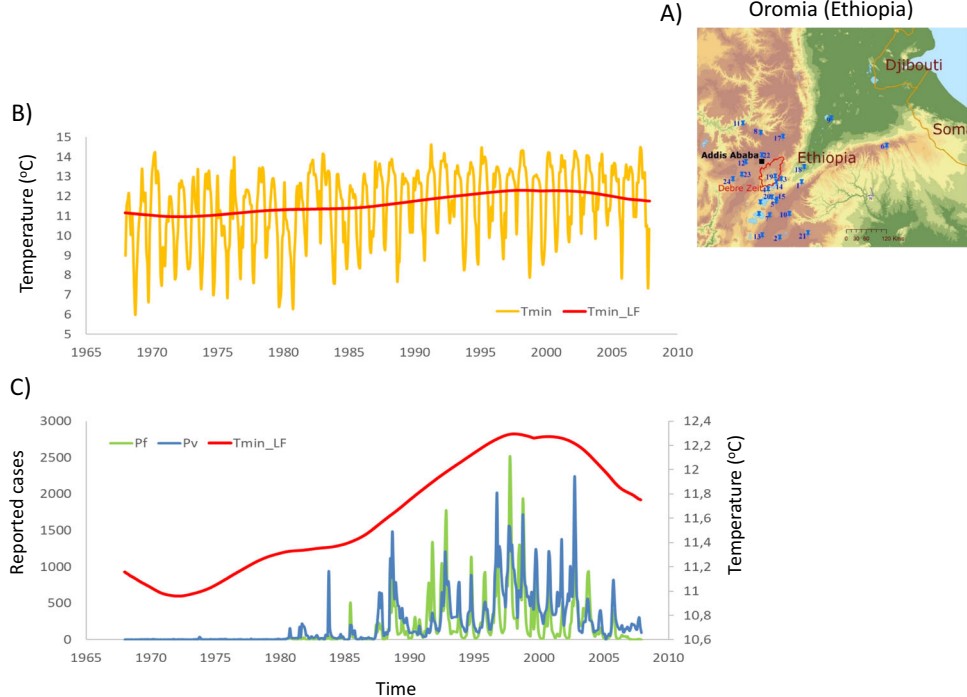

**Fig. 1 Geographical setting of the Oromia region (Ethiopia) and epidemiological evolution of malaria incidence since 1968.** Elevation of the region ranges from 1600 to 2500 m above sea level from the South East to the North West respectively. **A** The map shows the respective locations of Oromia in Africa, the 24 weather stations used in this study ("Methods" and Supplementary Information), and the malaria surveillance region in Debre Zeit (red contour). **B** Evolution in minimum temperature ($T_{min}$, yellow) and the long-term mean obtained by SSA ($T_{min\_LF}$; red line) for the interval 1968–2007 (this reconstructed component was significant at $p < 0.001$ against a red-noise null model; see "Methods" for technical details). **C** The malaria time series for both *Plasmodium vivax* ($P_v$, blue line) and *Plasmodium falciparum* ($P_f$, green line) correspond to confirmed malaria cases starting in January 1968. These data are shown together with the long-term composite minimum temperatures shown in **B**, an average from the 11 meteorological stations closest to the malaria surveillance area ("Methods" and Supplementary Information). There are two transmission seasons following the short (February–May) and long (June–August) rains. The main transmission season takes place from the beginning of September. (The regional map was produced with the ArcGIS software package, with an overlay of the GTOP30 digital elevation layer at 1 km × 1 km resolution obtained from USGS[86]).

September. The malaria time series for both parasites correspond to confirmed monthly cases starting in January 1968 from surveillance efforts in the entire Debre Zeit sector ("Methods" section, Fig. 1C). Outbreaks become evident at the end of the 1970s, increase in size towards the end of the 1990s, and are followed by an overall decrease in cases, with interannual variability overlaid on these general trends (Fig. 1C). The long-term variability for $T_{min}$ from Fig. 1B (red line) is overlaid with the evolution of malaria cases for comparison (Fig. 1C).

The concordant long-term patterns of variation between temperature and malaria suggest an important role of this climate factor in the turnaround of epidemic size at the beginning of the 2000s (Fig. 1; correlations $r > 0.95$, $p < 0.01$, Supplementary Table I). To address this role, we separated the $P_f$ time series into "training" and "prediction" sets including the reported cases pre and post 2000, respectively (Fig. 2 and "Methods" section). The training set was used to fit a dynamical model for the transmission of the disease by likelihood maximization; the prediction set, to compare the observed cases to the predicted ones generated with the best model driven solely on the basis of temperature (Fig. 2B, Supplementary Fig. 3 and "Methods" section). Specifically, we asked with the fitted model what would have been the post 2000 dynamics of the malaria system based solely on temperature changes and with no consideration of the strong interventions introduced in the mid-2000s. We are not assuming that malaria transmission occurred in the complete absence of interventions until this time. We are asking what would have been the effect of the observed variation in temperature if everything else

had remained the same than in the 1980s–1990s, including previous levels of intervention. The estimated value of the transmission rate ($\beta(t)$ in "Methods") including dynamical noise necessarily incorporates implicitly any effect of these earlier but limited control measures.

In the model, mean monthly temperature is used as an explicit, standalone input of the transmission rate at interannual time scales. We constructed the temperature covariate based on a clear empirical relationship between the total cases in each of the two transmission seasons and the average monthly temperatures over the respective rainy season that precedes each of these (see "Methods" and associated Supplementary Fig. 4 for details). The Maximum Likelihood Estimates (MLE) for the parameters with their well-defined confidence intervals are given in Supplementary Fig. 3 and Table included therein. Among these, the two temperature coefficients ($b_{T4}$ and $b_{T6}$) indicate a positive effect of this climate factor on the transmission for both seasons, significant and stronger for the second peak which is also the main malaria season. Figure 2B then shows results from an ensemble of predicted trajectories from numerical simulations of the model with the MLE parameters and for initial conditions sampled from the estimated state of the system for January 2000 ("Methods" section). The predicted size of seasonal epidemics reverses its trend at the beginning of this new decade, and mean monthly cases closely follow the observations up to 2004. Thus, the changes in temperature forcing in the context of existing, low, levels of control, have influenced the slowdown in reported cases at the beginning of the decade. Predictions and observations

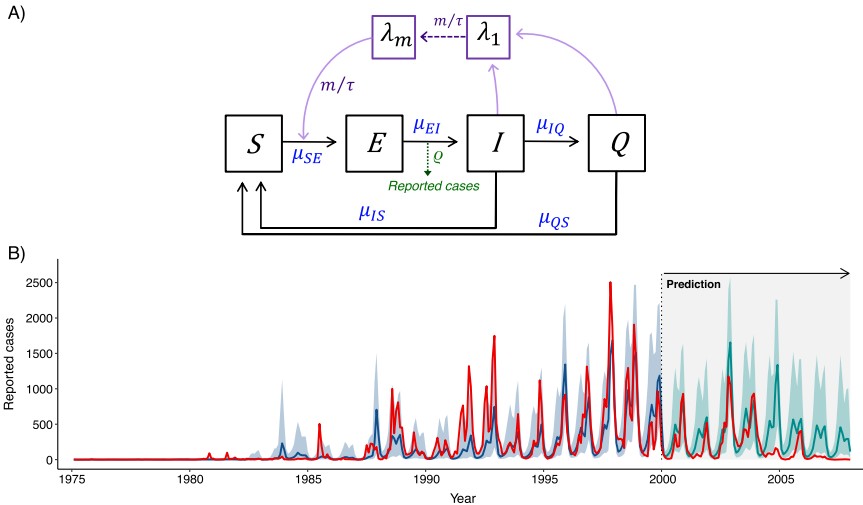

**Fig. 2 Stochastic transmission model. A** Diagram of the compartmental structure of the model. The model divides the human population into classes, for susceptible (S), exposed (E), infected, and infectious (I) individuals. An additional class is introduced, Q, for asymptomatic individuals that still carry the parasite after the initial infection but transmit it to the mosquito vector at a lower rate[77,78]. A second part of the model represents in a phenomenological way the mosquito component. As previously described[4,67], a chain of compartments ($\lambda_1,...\lambda_m$) effectively implements a distributed time delay between infections in humans and the force of infection (the per-capita rate of infection) experienced by a susceptible individual. This phenomenological delay represents the joint effect of the development of the parasite within the mosquito vector and the survival of this vector. The compartment chain generates a time delay in the form of a gamma distribution, a flexible form which by contrast to the more typical exponential distribution allows for the consideration of a mode or characteristic timescale. The effect of temperature is included in the transition rate representing transmission $\mu_{SE}$. The system of stochastic differential equations is given in the "Methods" together with the description of the measurement model (for cases reported with an error) and the way the temperature covariate enters in the transmission rate. **B** Comparison of predicted and observed falciparum malaria cases. The reported cases are shown in red. Median simulated cases (hindcasts) with the best model for the MLE (Maximum Likelihood Estimate) parameters are shown in blue for the time period of the training set data together with their uncertainty (shaded, for the 10% and 90% quantiles). Median predictions for the "out-of-fit" period are shown in green, also with their corresponding uncertainty. The Maximum Likelihood Estimates (MLE) for the parameters with their well-defined confidence intervals are given in Supplementary Fig. 3 and Supplementary Table included therein.

started diverging in 2004, as expected from the timing of the enhanced malaria interventions in the region[7] ("Methods" section).

We also fitted a nested model in which the incubation period in the human host is now fixed (to 12 days, a duration that is longer than our estimate, and closer to what is known about *Plasmodium falciparum* infections). Although we already know from the profile likelihood of this parameter (Supplementary Fig. 3) that its MLE value is shorter than 12 days, we address here whether our results are robust to fix this parameter at this longer value. Numerical simulations of the model with the fixed incubation period exhibit similar temporal behavior in the predictions driven by temperature (compare Fig. 2 with Supplementary Fig. 5). Thus, this parameter value is not critical to the conclusion that changes in temperatures predict the observed decline in the epidemic size of the early 2000s. An explanation for the shorter incubation period is that the model relies on this parameter together with the distributed delay in the force of infection (representing the vector implicitly) to generate an effective dynamical delay between the estimated seasonal transmission rate and the exact timing of epidemic growth and peak cases. Similar general results on predictions are obtained for a model whose covariates are based on minimum temperatures ($T_{min}$) (Supplementary Figs. 7 and 8).

To further investigate the coherency between malaria and temperatures, and to do so as a function of timescale, we decomposed the respective time series into separate (orthogonal) temporal components with Singular Spectrum Analysis (SSA[36–38], a method especially well-suited for this purpose in nonlinear systems). SSA reveals the existence of variability components at seasonal "high" frequency (hereafter, HF), interannual (IA), and interdecadal low-frequency (LF) scales, rather than a real trend

("Methods" section; Figs. 1C, 3, and 4). The reconstructed components correspond to respective periodicities shorter than (or equal to) one year for HF, between 1 and 10 years for IA, and longer than (or equal to) 10 years for LF.

High-frequency, seasonal, variability is of particular relevance to transmission since the seasonal cycle of both rainfall and temperature influences the abundance of the mosquito vector. Both variability components, HF and LF, exhibit highly concomitant changes among temperature and malaria cases for both parasites, especially for $T_{min}$ but also rainfall (Figs. 1C, 3A–C; correlations shown in Supplementary Table I). In particular, the reconstructed LF component for $T_{min}$ strongly covaries with those for both $P_v$ and $P_f$ malaria (Fig. 3A; see also Supplementary Table I). Moreover, from 1998 onwards there is a significant decrease in reported malaria cases for both parasites, with a concordant dip around the year 2000, despite their different epidemiology which includes the occurrence of relapses in $P_v$. A pronounced minimum in the 2000s is also apparent for the high-frequency seasonal components of $P_v$ and $P_f$ as well as for seasonal $T_{min}$ and precipitation (Fig. 3C, b). Moreover, the seasonal variability of incidence for both parasites (Fig. 3C) exhibits a low-frequency modulation (the envelope of the seasonal cycles) that tracks closely the trend in $T_{min}$ (Fig. 3A). In other words, the amplitude of the seasonality in incidence increases throughout the 1990s and decreases at the turn of the century. This time also coincides with a decrease in the amplitude of climate seasonality (Fig. 3C; time window shaded in blue, Fig. 3).

Analyses so far focused on regional climate. To develop a more comprehensive view beyond seasonality, we considered next connections to inter-hemispheric climate variability, to both the El Niño Southern Oscillation (ENSO) and the Pacific Decadal Oscillation (PDO). As mentioned before, changes in these climate

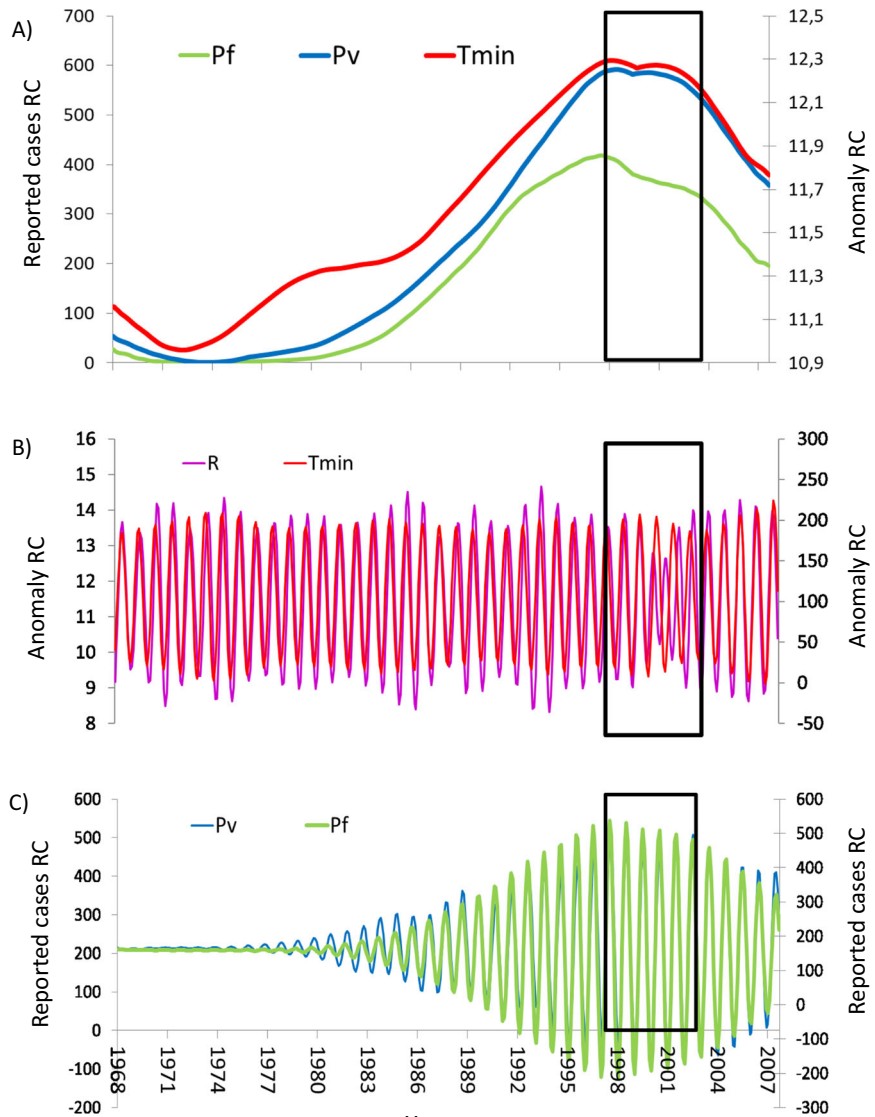

**Fig. 3 Comparison of the temporal coevolution of malaria cases, regional temperatures and rainfall, and remote climate drivers, for different components (low-frequency and seasonal). A** Low-frequency changes in $P_f$ cases, $P_v$ cases, and $T_{min}$ (the composite time series from 11 stations, $DZ_{reg}$), show concomitant variation and the existence of a slowdown in both malaria and temperatures. **B** The respective seasonal components of rainfall (R) and minimum temperature ($T_{min}$ $DZ_{reg}$) covary. Note the dip in both variables around the turn of the century and the concordance of the multi-annual envelope. **C** The respective seasonal components of $P_v$ and $P_f$ cases also covary, including the reduction in amplitude for both variables around the turn of the century. Note the long-term envelope of the seasonal cycles and its concordance with the trend of $T_{min}$ in **A**. The time period indicated by the black box corresponds to the years at the turn of the century, the focus of our investigation. All components shown were significant at $p < 0.005$ against both white noise and a red-noise null model (see "Methods" for technical details).

phenomena were implicated in the heat redistribution of the oceans underlying the apparent slowdown in global warming[26,32,33]. Moreover, both ENSO and the PDO are known to influence the seasonal migration of the Intertropical Convergence Zone (ITCZ), and ultimately the seasonal-to-interannual variability in rainfall and the variation in temperatures in eastern Africa[39]. We specifically addressed here whether these large-scale climate controls known to operate over the East Africa region are also associated with disease patterns and with regional temperatures.

At the interannual timescale, the reconstructed orthogonal components obtained with SSA ("Methods" section) for both $P_f$, $P_v$, and $T_{min}$ show clear covariation, with a temporary amplification of their magnitude in the late eighties and nineties, followed by a contraction afterward (Fig. 4A). Similarly, covariation

is seen for the reconstructed interannual component of the PDO (Fig. 4B). Thus, all interannual components in these panels concerning both disease and climate exhibit an increase in intensity after the 1980s and until the turn of the twenty-first century (Fig. 4). The impact of the large El Niño events of 1997/8, 2002, and 2006 on both regional climate and malaria cases in DZ is evident at both seasonal and interannual time scales (Figs. 3B, C and 4A, B shaded areas). In addition, the interannual component of the PDO appears to strongly covary also with those of the two malaria time series, even though it is not highly correlated with that of ENSO for the analyzed interval of 1968–2007 (correlation $r = 0.388$; $p > 0.05$). Taken together, these results suggest that large-scale climate variability exerts strong control over the region, in agreement with the more general literature on climate modulators of East Africa[40,41].

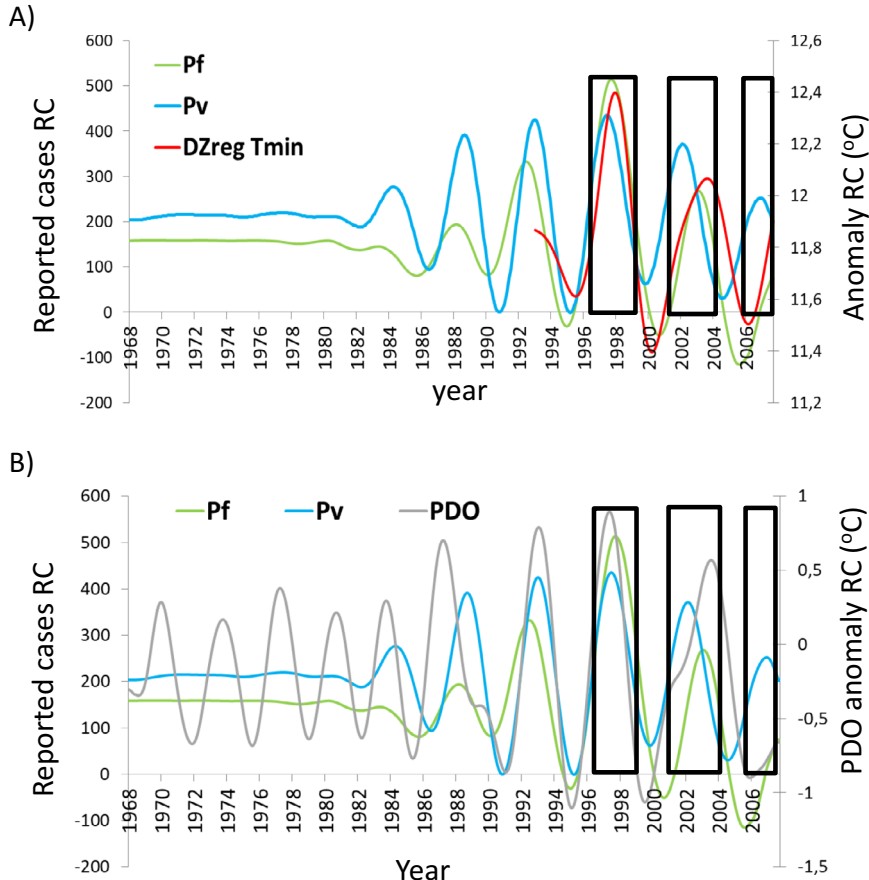

**Fig. 4 Interannual coevolution of malaria cases and remote climate drivers. A** The interannual components of $P_f$, $P_v$ cases, and $T_{min}$ ($DZ_{reg}$) are shown to covary in amplitude at these multi-annual temporal scales. **B** Similarly, the variability of $P_f$ cases and $P_v$ cases covaries with that of the PDO (Pacific Decadal Oscillation) index. Black boxes denote the three strongest El Niño episodes covered (1997/98, 2002, and 2006). In all cases, the variability for the entire interval 1968-2008 was decomposed by means of Singular Spectrum Analysis (SSA[36-38]) under the same analytical conditions. See "Methods" for details on both index construction and analyses. All shown components were significant against both white noise and a red-noise null model ($p < 0.001$).

The global modulators of regional climate and ultimately of malaria incidence in Oromia can be further confirmed by tracing spatially their association with the strong 1997/1998 malaria epidemics (Fig. 1A). To this end, we applied scale-dependent correlation analysis (SDC[42,43], "Methods" section) to the relationship between reported cases of $P_f$ and $P_v$ malaria and global sea surface temperature anomalies (SSTa). This statistical method is specifically formulated to analyze transient correlations between time series. Its spatial version considers these transient temporal associations between the time series for a given variable and those for a second variable at different locations in space ("Methods" section). Here, we specifically identify for which locations of a global grid, SSTa exhibit significant correlations with the disease time series in the given window of time (the 1997/1998 years). These local correlations highlight grid points in space of high and significant association in time. In particular, when an ensemble of the grid points with high and significant correlations to malaria cases constitute a coherent, contiguous, spatial area, its correspondence with an ocean region known to drive global climate variability supports an influence of the associated climate phenomena on disease dynamics (e.g., regions of the Pacific Ocean associated with ENSO or the PDO). Figure 5A–D shows the identified SSTa regions associated with the strong 1997/1998 malaria epidemics for different lead times of up to 4 months before the malaria peak. The significant positive correlations clearly identify regions of the Pacific Ocean

indicative of both a strong warm PDO phase and warm EN conditions. Both climatic anomalies have large positive phases in 1997/98 producing synergistic effects and extreme impacts on the regional climate of Ethiopia, and therefore on local malaria epidemiology. Specifically, an anomaly of over 2.5–3 °C in regional SST over the Ethiopian coast occurred under the strongest EN on record in 1997, concomitant to a positive PDO phase (Fig. 5). Similar correlation patterns to those of $P_f$ and SSTa were obtained for this same EN episode when considering for the climate grid, surface air temperatures at two meters height (or $T_{2m}$) with anomalies reaching 3 °C (Supplementary Fig. 10). The similarity in the spatial correlation patterns further supports the link between the dynamics in the Pacific Ocean and climate in East Africa, as no pre-determined ENSO frequency is inherent in the air temperature except following strong EN events.

The effect of EN can vary however across events, with only particular events exerting a clear influence on the region of interest and others not (Supplementary Figs. 9 and 11). SDC maps between the Niño3.4 time series and the global SSTa grid show this nonlinear relationship between El Niño conditions and ocean temperatures, especially for the western Indian Ocean (WIO). Of the 11 EN events within the study interval, only the EN events of 1997, 1972, 1982, and 1968 are associated with large anomalies in the WIO and Ethiopia (Supplementary Fig 11).

We left out of this study the more local and higher-frequency roles of Indian Ocean modes of variability such as the Indian

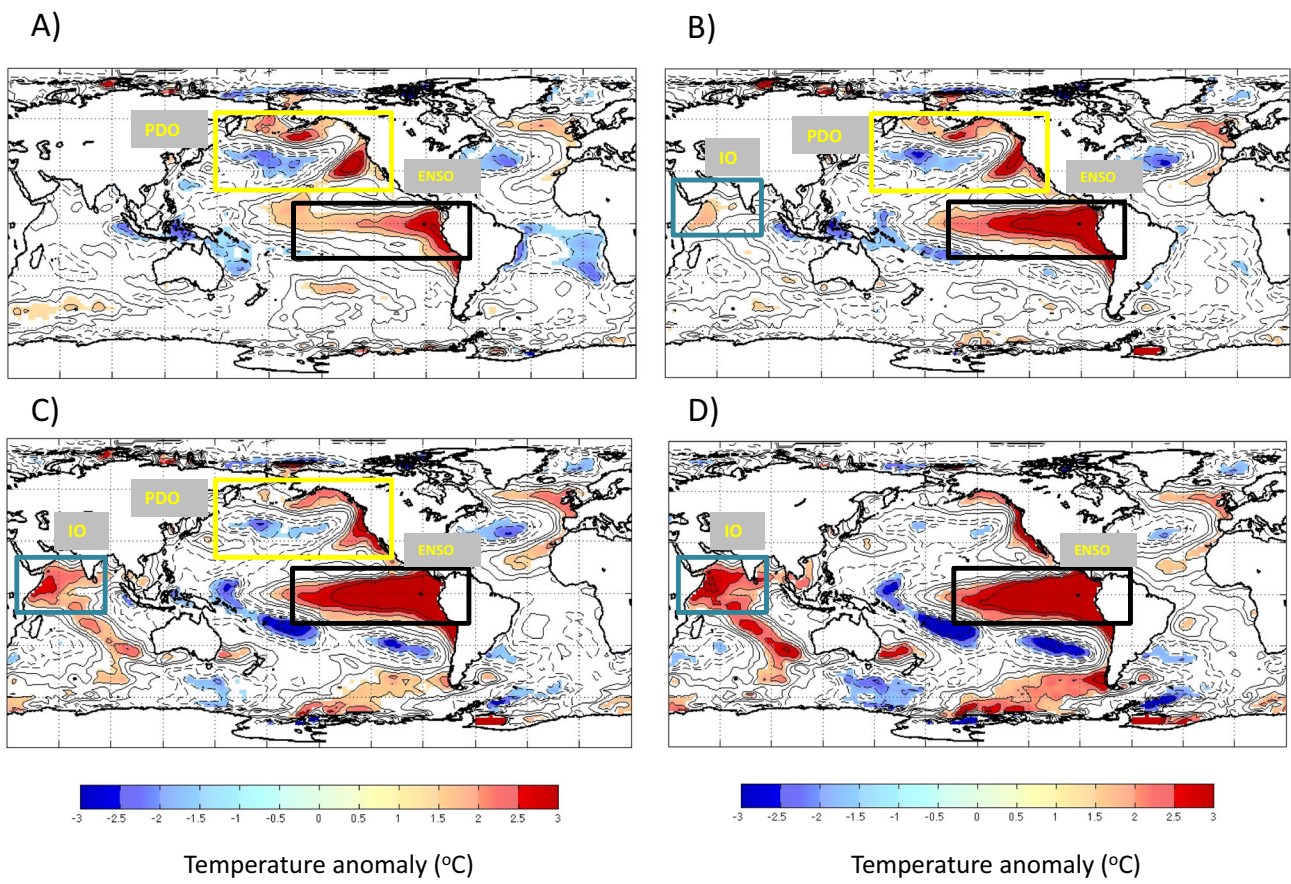

**Fig. 5 Correlation maps showing the signature of *Plasmodium falciparum* malaria in Oromia (Ethiopia) onto values of Sea Surface Temperatures for the Pacific regions of El Niño (EN) and the Pacific Decadal Oscillation (PDO).** SDC[75,76] spatial analyses are shown for the main malaria peak in the record in 1997/98, coinciding with the strongest EN event. Correlations are computed between the malaria time series and the SSTa time series at each point of the global ocean. Grid points with a significant correlation ($p < 0.01$) are colored. Regions of significant correlation are found within the boxes that delimit the PDO and EN regions of the Pacific, as well as the Indian Ocean (IO). The panels correspond to different months leading (−) or lagging (+) the malaria peak month in 1997/98, and show the developing and significant SST anomalies in the global ocean that are correlated with malaria incidence. SSTa, respectively, leads the malaria peak by 4 months in **A** and two months in **B**; they exhibit no lag in **C** and lag behind the peak by two months in **D**.

Dipole Mode (IDM)[44], as they are for this time interval tightly linked to ENSO control and do not exert a comparable modulation over the interannual or decadal time scales (Supplementary Fig. 12). A clear covariation between the IDM and the Niño3.4 indices is evident (Supplementary Fig. 12A), with higher-frequency components being characteristic of the IDM and no relevant decadal variability structure (Supplementary Fig. 12B, C).

At interannual time scales, changes in global mean surface temperatures should reflect variability in ENSO and the PDO, given the control exerted by these phenomena on the world's largest ocean basin. Indeed, this relationship between global and regional variability is clearly apparent when the LF trend component is removed from the GMST (HadCRUT4 anomaly in Supplementary Fig. 13A). The remaining variability in the GMST after removing the trend exhibits close covariation with the PDO (Supplementary Fig. 13B) and mostly ENSO (Supplementary Fig. 13C).

As good station coverage is far from optimal in Africa for the time interval of interest, and in particular, for the earlier part of it, we relied on an atmospheric climate model to properly simulate the effects of ENSO and the PDO on regional climate. We examined the consensus atmospheric model simulations for the region with the Community Atmosphere Model version 5 (CAM5[45]). CAM5 includes recent physical parameterization enhancements to realistically simulate

full aerosol-cloud interactions, (http://www.cesm.ucar.edu/models/cesm1.0/cam/docs/description/cam5_desc.pdf). We used the atmospheric model intercomparison project (AMIP) simulations with settings selected so that the generated SSTs retain interannual and short-term decadal variability. For this purpose, the long-term trend was not included, and therefore cannot interfere with the temporal scales we wish to investigate. Correlations of the sea and land surface temperatures with Niño3.4 in the two-time intervals of interest (1979–2000 and 2001–2016), before and after the slowdown respectively, support a strong dynamical role of both ENSO (Supplementary Fig. 14) and the PDO (Supplementary Fig. 15) in modulating climate over East Africa and the nearby Indian Ocean, which changes at the turn of the century. The influence of ENSO appears to weaken, whereas that of the PDO strengthens, around the turn of the century, in close correspondence to the association patterns obtained in the previous time-series analyses of the data.

## Discussion

Decadal variability in malaria cases for both parasites, *P. falciparum* and *P. vivax*, tracks the trends in regional temperatures closely. We have shown that changes in local temperatures can explain what we could also call the "slowdown" in malaria epidemics. For the region of Debra Zeit in Ethiopia, a transmission

model driven by changes in temperature and under counter-factual conditions of no enhanced public health intervention, accurately predicts the observed decline in epidemic size post 2000. Notably, this decline precedes the known timing of enhanced control efforts. Thus, changes in temperature have facilitated the effect of control, working synergistically with it, in the sense of precluding the large epidemics of the 1990s. This conclusion is further supported by the statistical analyses of time series, showing the coincident patterns of variation of both falciparum and vivax cases with temperature (as well as rainfall) at the multiple temporal scales considered, from seasonal to inter-annual to decadal.

Thus, the results of our transmission model indicate that changes in temperature could have driven, fully or in part, the reversal from an increasing to a decreasing trend in seasonal epidemic size. Malaria dynamics and their response to climate change worldwide are of course complex phenomena involving many other drivers, including land-use, migration, and socio-economic conditions[7]. As designed, our modeling analysis isolates the effect of temperature, all else remaining constant. An alternative hypothesis other than the temperature would involve changes in the frequency of drug resistance[46]. The co-occurrence of both parasites in Ethiopia already allowed us to dismiss this explanation, as only *P. falciparum* and not *P. vivax* exhibited resistance to chloroquine in the 1980s and 1990s[7]. The fact that in this region incidence by both parasites covaries in similar ways to temperature (Fig. 3) thus rules out drug resistance as the causative factor behind the trends. In addition, drug resistance would need to first increase and then decrease in frequency exactly at the same time than both incidence and temperature. Increases in the number of reporting clinics are another possible confounding factor. These would imply however a change in the total population covered, which was taken into account by incorporating observed population size explicitly in the transmission model.

Our modeling approach does not imply that control efforts were completely absent before the turn of the century. Effects of control can be implicit in the estimated transmission rate and dynamical noise. We are asking what would have been the predicted trajectory of cases if the control level had not changed. In fact, during the pre-2005 years leading to the large epidemic in 2003, which affected more than 200 districts[47], malaria control measures had been weak overall with diagnostics facilities only available in few government-owned malaria sectors. Starting with the policy change to use ACT for treatment of *P. falciparum*[48], the beginning of health extension workers assigned at sub-district level providing diagnostic and treatment services[49], as well as the large-scale distributions of LLINs and IRS across all regions[50], represented a significant shift both in the local and national responses toward the control of the disease in 2005. The Debre Zeit malaria diagnosis and treatment facility and four other facilities in the larger East Shoa zone were the most recognized one-window centers, solely providing malaria examination and treatment services in the area[51].

Although the results of the time-series analyses are similar for both parasites, the transmission model was developed and parameterized for *P. falciparum* to avoid the complexity of relapsing infections characteristic of *P. vivax*. Extensions could consider a similar analysis with a model for vivax malaria[52]. We modeled spatially aggregated malaria cases. Our process-based transmission model could be improved further by explicitly considering space, here elevation. Interestingly, predictions from the fitted temporal model reproduce the data extremely well after the large epidemics of 1997–1998, coincident with a large El Niño. Previous work has shown that during the El Niño events of the 1990s, the transmission of the disease expanded upwards in elevation, and did not contract after 1997/1998[7]. This observation

suggests that a purely temporal model is best able to capture transmission dynamics once this altitudinal distribution has expanded.

To consider large-scale, global, drivers of climate and their role in the region of interest, including malaria dynamics, we concentrated here on ENSO and the PDO, two major phenomena in the climate variability of the Pacific Ocean. As mentioned earlier, this choice stems not only from the important role of the oceans in the dynamics of global warming but also from the evidence involving the large El Niño event of 1997/98 and the La Niña of 1999/2000 in the so-called slowdown in warming at the turn of the century[31]. As pointed out in the IPCC AR5 report, 90% of the heat resulting from global warming during the last four decades has been accumulated in the oceans, and the periodic occurrence of El Niño can act as a vent to exchange this heat from the ocean to the atmosphere[53]. Partially transferred to the Indian Ocean via the Indonesian throughflow and a modified Walker atmospheric circulation, this heat is reflected in the warming trend over that region[31]. As the IPCC AR5 also remarks, the warming trend over the Indian Ocean is a major contributor to, and largely in phase with, the overall trend in the global mean SST[53,54].

We did not analyze here direct associations between a time series of global temperatures and malaria (or regional climate) in Ethiopia, as global trends measured in that way reflect an emergent average from variation at multiple temporal and spatial scales, rather than a tangible driver related to the dynamic processes that influence the region of interest. We provided instead evidence for a chain of effects from global drivers of climate variability to regional climate variability in East Africa to local malaria in Ethiopia. A weakening of the atmospheric forcing by El Niño is evident post 2000, and further research is needed to interpret the enhanced appearance of La Niña events in the following decade. Conversely, the opposite is observed for the PDO, with a reversal towards a stronger association, even though the length of the time span considered limits conclusions on this mode.

Previous studies have reported significant correlations between Pacific SSTs and Ethiopian climate, with the mechanisms behind the role of ENSO still poorly understood (but see Diro et al.[55]). The main focus has been on rainfall rather than temperatures, on drought conditions during El Niño years, and a downward trend in the Long Rains over parts of the Great Horn of Africa starting at the end of the 1990s[55–61]. Our complementary attention to temperature stems from its dominant role as the limiting climate factor of malaria transmission in highlands, with rainfall influencing mostly the timing of the seasons.

A large body of climate literature has also addressed linkages at longer time scales between the Pacific Ocean and the East Africa region, including the Western Indian Ocean (WIO). For example, Hoell et al.[62] suggested that human-induced changes in tropical SST directly exacerbated the effects of natural Pacific interannual and decadal variability, enhancing both warming and drying in the Great Horn of Africa. Roxy et al.[54] provided evidence for the direct influence of greenhouse warming on the region of interest. They proposed that the long-term warming trend over the western Indian Ocean is highly dependent on the asymmetry of the ENSO teleconnection, with El Niño leading to warming but La Niña not inducing the opposite cooling. They further suggested an increase in the frequency of El Niños in the interval of 1950–2012, which would have impacted the WIO, and noted an intensification of La Niñas following strong El Niño events (e.g., EN in 1997/98 and LN in 1999/2000). Interestingly, the more recent cool conditions over the eastern Pacific might be due to feedback from a warmer Indian Ocean[54] bringing the sequence of cause and effects into a complex and vicious cycle that requires further research. Several other studies have also noted that

warming trends over the Indian and Atlantic Oceans lead to La Niña-like conditions over the Pacific[63–65]. The interconnectedness of the different ocean basins in the tropics is increasingly appreciated, including negative feedbacks from warmer Atlantic and Indian oceans to the eastern Pacific, that may have played a role in the warming hiatus and remain to be further elucidated[31].

Our results should motivate similar analyses for other highlands in East Africa, including those in Kenya where long-term surveillance programs at two tea plantations have been the subject of a number of studies on increasing trends in incidence from the 1980s to the 1990s (reviewed in refs. [48,66]). These extensions will require the identification and collation of long-term epidemiological records where they exist, and of information on the implementation of control measures over time as well. A previous study[66] already raised the question of whether factors other than heightened intervention in the form of large-scale bed net distribution and improved case management, might explain observed decreases in reported infections and mosquito abundances in sub-Saharan Africa. The study reports mosquito data in two endemic low-altitude locations of NE Tanzania for two respective periods of time, from 1998 to 2001 and from 2003 to 2009. The role of climate factors in these very different epidemiological conditions cannot be extrapolated from our study, and cannot yet be fully addressed with such short periods, especially for questions on trends and their changes. Consideration of longer data sets on mosquitoes and infections in endemic regions is warranted, including the role of factors other than climate such as urbanization and land-use change.

The evidence for the slowdown in warming influencing malaria transmission demonstrates the strength of the coupling between the disease and climate. This temporary slowdown acting synergistically with the enhanced control efforts emphasizes the relevance of considering climate conditions when evaluating and planning public health intervention in epidemic regions in general. An apparently persistent role of climate conditions in 2006, after strong control had already been initiated, further underscores this point. The manifestation of climate effects at multiple temporal scales should be expected. Relatively short multi-annual trends in improved health are not sufficient to conclude persistent effects of intervention, which would entail the danger of complacency and relaxation of these efforts[67]. Although considerable progress has been made in regions of low malaria transmission including the highlands of Ethiopia[6,7], consideration of underlying and predicted climate conditions should inform changes and investment in public health plans. In the absence of regional elimination, an enhanced dialog between climate science and public health should allow better consideration of the environmental context in which the fight against the disease is being conducted. These conclusions are especially relevant under the current targets of malaria elimination, the successes of the last ten to fifteen years mainly in epidemic, low transmission, regions, and the increased realization of *P. vivax* infections as a neglected public health problem in Ethiopia[50,51], and beyond 2015[52].

## Methods
The malaria study region is described in detail elsewhere[7]. The reported cases for both $P_f$ and $P_v$ were confirmed through microscopy examination of blood slides from clinical (febrile) cases seeking diagnosis and treatment. They were provided by the government malaria center in Debre Zeit, established in the late 1960s to aid Ethiopia's eradication efforts. For consistency, cases diagnosed in government clinics outside the original reporting sector between 1993 and 2007 were not included, and cases from four new clinics opened after 1993 in Debre Zeit town, Mojo, and Chefe Donsa were added. The malaria cases post 1993 are originally reported spatially for 159 administrative units known as kebeles in Debre Zeit[7] and aggregated here to obtain monthly totals and construct a time series from 1968 for a consistent area. Systematic control efforts were applied in the region starting in September 2005, including the introduction of the new drug treatment (ACTs) for $P_f$, and a significant increase in vector control (indoor residual spraying, IRS) and

prevention (Insecticide Treated Bednets, ITNs) after the epidemic seasons of 2002–2003. While we cannot control for some variation in the coverage of the surveillance system up to 2004, there was no structural change in terms of health care policy or access to health care that could lead to significant inconsistencies or trends over this period of time. Changes to the treatment policy for *P. falciparum* (from SP to ACT) only took effect beginning in the fourth quarter of 2005[48], and could not, therefore, have an effect on the time period we used to inform our dynamical model. Inclusion of cases from other neighboring malaria treatment facilities (four in the East Shoaled Zone) which treated patients traveling from the Debre Zeit sector, ensures that we captured all cases regardless of which facility they happened to visit.

Monthly averages of daily minimum and maximum temperature data and precipitation for stations of the Oromia region were obtained from the Ethiopian National Meteorological Agency (NMA). We generated average time series for these variables from the 11 stations closest to the study site (Supplementary Note 1). Compositing several time series provides a more robust estimate and eliminates local inconsistencies. In particular, the Debra Zeit station which is the closest to the region of malaria surveillance exhibits a pronounced dip for $T_{min}$ in 2004, coincident with the movement of the station to a higher elevation. This particular station was therefore excluded from the chosen ensemble of 11. To validate the use of the resulting average time series, these were compared in both the frequency and time domains with the corresponding composite average of all the 24 weather stations in the Oromia region, as well as to the previous four-station average[7] (Supplementary Note 1 and Supplementary Figs. 1 and 2).

To compute both global and regional correlations, reanalysis data sets were used for global land surface temperature and sea surface temperature (Climate Research Unit TS4.01, at 0.5-degree spatial resolution[68]; NCEP/NCAR reanalysis[69,70]). Climate reanalysis combines past observations with models to generate consistent time series of multiple climate variables. Reanalysis products are used extensively in climate research and services, including for monitoring and comparing current climate conditions with those of the past, identifying the causes of climate variations and change, and preparing climate predictions.

To further investigate linkages between East African climate and the Pacific Ocean basin dynamics, we used both the El Niño index (Niño 3.4, defined as the temperature time series within the box 5_N–5_S, 170_W–120_W from NOAA-ERSST-V3 and NOAA-OISST-V2, www.esrl.noaa.gov/psd/), and the Pacific Decadal Oscillation index (PDO, defined as the leading principal component of North Pacific monthly sea surface temperature variability, poleward of 20N[71]).

Singular spectrum analysis (SSA[36–38]) was applied to separate different orthogonal components in both the malaria time series and the regional climate ones. SSA involves the spectral decomposition (eigenvalues and corresponding eigenvectors) of a covariance matrix obtained by lagging the time series data for a prescribed number of lags M called the embedding dimension. There are two crucial steps in this analysis for which there are no formal results but useful rules of thumb: one is the choice of M; the other is the grouping of the eigenvectors to define the specific major components and reconstruct them. Typically, the grouping of the eigencomponents is based on the similarity and magnitudes of the eigenvalues, their power (variance of the data they account for), and the peak frequency of resulting reconstructed components (RC). For the selection of the embedding dimension one general strategy is to choose it so that at least one period of the lowest frequency component of interest can be identified, that is M > fs/fr, where fs is the sampling rate and fr is the minimum frequency. Another strategy is that M be large enough so that the M-lagged vector incorporates the temporal scale of the time series that is of interest. The larger the M, the more detailed the resulting decomposition of the signal. In particular, the most detailed decomposition is achieved when the embedding dimension is approximately equal to half of the total signal length. A compromise must be reached, however, as a large M implies increased computation, and too large a value may produce a mixing of components.

Specifically, it is well-known that to characterize interannual variability (in between seasonality and low-frequency components), as in the case of ENSO, it is convenient to use filters that allow to properly characterize -and separate- components at the quasi-biennial and the quasi-quadrennial interannual ENSO scales[42]. Following customary approaches in climate research, and to achieve proper separation, we used a scale $M = 40$, in between these two main periodicities[42,43].

Once the main reconstructed components were generated with SSA, we applied spectral analysis with both the Maximum Entropy Method (MEM) and the Multi-Taper method (MTM), to identify the dominant frequencies in their power spectrum[72–74]. MEM is especially aimed at short and noisy time series, and MTM is a flexible nonparametric method that reduces the variance of spectral estimates by using a small set of tapers. In this way, we extracted the different variability components of each time series and compared data sets (e.g., IDM and ENSO, with only the latter exhibiting a clear period in the decadal range, namely 12.5 years; Supplementary Fig. 12).

To analyze correlations that are local in both time and space, we relied on the SDC Map methodology, which includes consideration of different time lags and implements significance tests based on permutations[43,75,76]. The correlation maps resulting from this method allow the identification of regions of the global ocean whose Sea Surface Temperatures exhibit significant correlations with malaria in Ethiopia at a given lead time.

A stochastic transmission model for malaria was developed that follows previous[77–79] extensions of the compartmental SEIR model. The diagram illustrating its compartmental structure for the human population and for the phenomenological representation of the mosquito is shown in Fig. 2A. Alternative formulations consider two susceptible classes, rather than a single one, to differentiate individuals who have acquired immunity from the previous infection and are asymptomatic upon secondary infection[79–81]. Here, consideration of a single class provides a simplification by not tracking explicitly the infection process in the partially immune individuals. For low transmission regions, this simplification precludes problems with parameter identifiability and captures the role of a reservoir of transmission played in the dynamics by asymptomatic individuals. We adopt here this simplification and assume that individuals in class Q are asymptomatic and not reported in the surveillance system. Such individuals should not make an important contribution to the force of infection in unstable regions; we include it here for completeness and let the data determine their importance and associated parameters. Previous work has indicated that a model that includes them better fits the data than a traditional SIRS formulation, even when their contribution to the overall force of infection is small[78]. The corresponding system of stochastic differential equations is given by:

$$\frac{dS}{dt} = \left(\delta P + \frac{dP}{dt}\right) + \mu_{IS}I + \mu_{QS}Q - \mu_{SE}S - \delta S \quad (1)$$

$$\frac{dE}{dt} = \mu_{SE}S - \mu_{EI}E - \delta E. \quad (2)$$

$$\frac{dI}{dt} = \mu_{EI}E - \mu_{IS}I - \mu_{IQ}I - \delta I \quad (3)$$

$$\frac{dQ}{dt} = \mu_{IQ}I - \mu_{QS}Q - \delta Q \quad (4)$$

The different parameters $\mu_{XY}$ denote transition rates between the classes X and Y. In particular, $\mu_{IS}$ and $\mu_{IQ}$ denote recovery from symptomatic infection, with and without the acquisition of immunity respectively, and $\mu_{QS}$ denotes both recovery from asymptomatic infection and loss of protection against clinical disease (i.e., a return to full susceptibility). Rate $\mu_{EI}$ corresponds to the transition from exposed to infectious, and therefore $\frac{1}{\mu_{EI}}$ is the average time of development of the parasite within the human host, or the average time from a host receiving an infectious bite to the parasite producing the blood stages transmissible to a biting vector. We consider that the influx of new susceptible individuals equals the human death rate $\delta$ plus the observed demographic growth of the population $\frac{dP}{dt}$ estimated from data. That is, we incorporate an influx of new susceptible individuals with the rate $\left(\delta P + \frac{dP}{dt}\right)$ to match the observed population from which the reported data comes from and its growth in the region.

To phenomenologically represent the effect of the vector, we numerically implement a distributed time delay in the force of infection as a gamma distribution (with mean $\tau$ and variance $\tau^2/m$, $m = 2$). Technically, this is done with a set of sequential transitions through a chain of identical $m$ stages between the "latent" force of infection $\lambda$ and its actual value $\mu_{SE}$ experienced by susceptible individuals. For $m = 2$,

$$\frac{d\lambda_1}{dt} = (\lambda - \lambda_1)2\tau^{-1} \quad (5)$$

$$\frac{d\lambda_2}{dt} = \frac{d\mu_{SE}}{dt} = (\lambda_1 - \lambda_2)2\tau^{-1} \quad (6)$$

As in refs. [62] and [64], we chose to consider only two stages in the chain, which is sufficient to generate a unimodal distribution. To complete the model and introduce the effects of climate covariates, we need to specify the latent force of infection $\lambda(t)$

$$\lambda(t) = \beta(t)\left(\frac{I_1 + qQ}{P(t)}\right) \quad (7)$$

where the transmission rate $\beta(t)$ incorporates seasonality, interannual effects of temperature in two critical windows of time preceding the two transmission seasons, and environmental noise. We write

$$\beta(t) = \exp\left[\sum_{k=1}^{6} b_k s_k + b_{T_4}s_4\text{TEMP}_1 + b_{T_6}s_6\text{TEMP}_2 + b_T\text{TEMP}\right]\frac{d\Gamma}{dt} \quad (8)$$

where the sum implements a flexible (nonparametric) seasonality through the coefficients ($b_k$) of an orthogonal basis of periodic b-splines, $s_k(t)$ $k = 1...,6$ (see Supplementary Fig. 6 for the shape of these functions). The two following terms overlay the interannual effects of temperature (via covariates TEMP$_1$ and TEMP$_2$) on this seasonal pattern by localizing these effects in two specific windows of time given by the 4th and 6th b-spline, respectively, while TEMP corresponds to a monthly temperature covariate whose value for a given month i is calculated as a moving average over months $i - 3$ to $i$, with the resulting time series normalized by subtracting its mean and dividing by its standard deviation. The final term in the expression for the transmission rate denotes environmental noise, modeled with a Gamma distribution $\Gamma$ to represent unaccounted variation beyond seasonality and interannual temperature forcing. Supplementary Fig. 6 also illustrates the deterministic part of the transmission rate $\beta(t)$ obtained from fitting the model to the time series of cases, for

the estimated coefficients $b_k$ (determining its seasonality) and $b_{T_4}$, $b_{T6}$ and $b_t$ (determining its interannual variation and trend).

The covariates TEMP$_1$ and TEMP$_2$ are obtained by averaging monthly temperatures over the windows of time (February to May for TEMP$_1$ and June to September for TEMP$_2$) preceding the two transmission seasons and corresponding to the short and long rains, respectively[7]. We further subtract a threshold value of 17 °C and make the covariates equal to zero below this threshold. This functional form was selected following the observation that accumulated cases during each of the respective transmission seasons is nil below, and increases with the temperature above, a clear threshold value of 17 °C (Supplementary Fig. 4). This threshold is consistent with the empirical range for the lower thermal limit of the basic reproductive number ($R_0$) of falciparum malaria, established from physiological measurements of the parasite in Anopheles mosquitoes in the laboratory[9].

To consider that cases are under-reported and measured with error, we introduce a measurement model given by a negative binomial distribution so that cases~Negbin ($\rho C_i, k_i$) with overdispersion $k_i$ and reporting rate $\rho$. The variable $C_i$ denotes the accumulated new infections (or incidence) sampled in our simulations from the transitions from E to I during a given interval of time (here, a month).

The model is fitted to the time series data for malaria cases between 1980 and 1999, with a sequential Monte Carlo method based on particle filtering known as MIF for Likelihood Maximization by Iterated Filtering implemented in the R-Package pomp[80,81]. This method allows consideration of both process and measurement noise, as well as the partial observation of the system. It is now widely applied to the study of population dynamics in infectious diseases with epidemiological models informed by time series.

Predictions from January 2000 forward were then generated by simulation of the "best" model (with the Maximum Likelihood Estimate, MLE, parameters). To specify the initial conditions, we also need an estimate of the full state of the system at the end of the training period (December 1999). These estimates (including their uncertainty intervals) are provided by the filtering algorithm and are used to initialize the simulations for the predictions. Because the model is stochastic, we generated 1000 simulations and obtained the median and 10–90% percentiles for the monthly cases from 2000 to 2008.

Climate records in Africa are sparse, often discontinuous with multiple gaps and inconsistencies. This is particularly the case for Ethiopia, where meteorological time series tend to be short and not properly curated. Climate models provide an alternative to address connections between distant regions, including atmospheric mechanisms behind these teleconnections. Our climate analyses were reinforced with atmospheric simulations from the atmospheric intercomparison project (AMIP) with the ESRL (NOAA/Earth System Research Laboratory)—Community Atmosphere Model version 5 (CAM5). CAM5 provides recent physical parameterization enhancements to better simulate full aerosol-cloud interactions[45], such as cloud-aerosol indirect radiative effects (http://www.cesm.ucar.edu/models/cesm1.0/cam/docs/description/cam5_desc.pdf).

To compare potential changes in the forcing by ENSO and the PDO of regional temperatures before and after the hiatus in 2000 (both SST and air temperatures), we used AMIP simulations with 1880s Radiative Forcing that include conditions in which SST has been detrended and adjusted to the 1880 equivalent mean conditions[82] (but retain interannual and decadal variability). Sea ice is set to a repeating seasonal cycle of roughly 1979–1990 (i.e., pre-emergence of the melt out)[83]. The greenhouse gases (GHG) and ozone concentrations are adjusted to their 1880 values. The applied Greenhouse Gases (GHG) are those of the CMIP5 recommendations (for annual average and global mean concentrations[83]), ozone from the AC&C/SPARC ozone database[84,85], and aerosols from CAM5 ECHAM5 (time-varying aerosol content, and volcano aerosols from 1979 to 2005). After 2005, RCP6 aerosols (with no volcanoes) are used. CAM5.0 incorporates a number of enhancements to the physics package (e.g., several adjustments to the deep convection algorithm), and collectively these improvements yield a significantly improved atmospheric modeling capability. The simulated temperature and SST fields were then correlated with the Niño3.4 and PDO indices for the periods of 1979–2000 and 2001–2016 separately. (Although the temporal span of the two-time intervals compared could impose a limitation, the results we obtained were aligned with those from the time series analyses based on data).

**Reporting summary**. Further information on research design is available in the Nature Research Reporting Summary linked to this article.

## Data availability

The input data for the atmospheric simulations are part of the customized initial and boundary conditions used in the simulations specified in the "Methods" section, and are all accessible from http://www.pa.op.dlr.de/CCMVal/AC&CSPARC_O3Database_CMIP5.html. Monthly averages of daily minimum and maximum temperature and precipitation for stations of the Oromia region were obtained from the Ethiopian National Meteorological Agency (NMA). These data can be found at https://github.com/pascualgroup/Malaria-highlands. The El Niño index (Niño 3.4) and the PDO index were obtained from www.esrl.noaa.gov/psd/. The reanalysis data can be found at https://crudata.uea.ac.uk/cru/data/hrg/. The epidemiological data are available from the authors upon request.

## Code availability

The code developed to fit the transmission model via iterated particle filtering (MIF) and to produce predictions with this model, using the R-package Pomp, is available at https://github.com/pascualgroup/Malaria-highlands. The code in Python for Scale-Dependent Correlation Analysis (SDC) developed by X.R. can be found at https://github.com/AlFontal/sdcpy.

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

## Acknowledgements

We express our sincere thanks to A. Mekuria, A. Yeshiwondem, D. Dengela, A. Haile-mariam, A. Woyessa, S. Chibsa, and the field and laboratory health workers in Ethiopia for their active involvement in the data collection and to the World Health Organization (WHO) Office for Ethiopia, Ministry of Health, and Center for National Health Development in Ethiopia for their technical support during the data collection. We are indebted to A. Nega Tulu for sharing the malaria data of Debre Zeit before 1993. We thank the Research Computing Center (RCC) of the University of Chicago for providing access to their cluster. We acknowledge support from the Spanish Ministry of Science and Innovation through the "Centro de Excelencia Severo Ochoa 2019-2023" Program (CEX2018-000806-S), and support from the Generalitat de Catalunya through the CERCA Program.

## Author contributions

M.P. and X.R. conceived the study and wrote the paper. X.R. conducted climate data processing, time series analyses, and atmospheric simulations. M.P. formulated the epidemiological model, and P.P.M. and M.P. implemented parameter estimation and predictions. A.S. provided malaria, demographic and temperature data, expertise on malaria in the region, and contributed to the figures. All authors discussed results and contributed to the final writing.

## Competing interests

The authors declare no competing interests.
