## [Peer Review File · Nature Communications]

Reviewers' Comments:

Reviewer #1:

Remarks to the Author:

I think the paper improved since the last round of revisions. My overall impression is that recent updates (related to former round of reviews) sometimes support the manuscript further, sometimes it can be quite the opposite. There are still points to clarify further before the manuscript can be accepted for Nature Communication, thus major revisions. See my comments in the following for further details.

Major comments

L124-127: paragraph commenting the match between climate variable and Pf/Pv cases at different time scales (seasonal vs interannual vs decadal). The authors mention a good match at all periodicities between climate variables and Pf/Pv cases – this statement is backed up by eyeballing time series on Figure 2. I think the authors should provide a more robust metric to assess this “match” further: simple correlations between climate and case RCs at all frequencies with associated significance levels could be provided somewhere in the manuscript (text and/or table in Supp Mat) to strengthen this point I think.

Figure 5: I think there is a serious methodological issue here. The authors correlate global SST data with SDC outputs for Pf falciparum cases over a 24 months window (in 1997-98 so during one of the largest El Nino on record). Thus, the correlation pattern obviously maximises an El Nino pattern (by construction! – a composite analysis might yield similar results as well). Importantly, the analysis is not conducted for the whole duration of the time series. If I had to do this properly, I will extract the interannual component of the Pf case data (already done here). Then I will filter out the low frequency component in the SST signal (to only retain the interannual variability) and then I will correlate them over the whole time period using lags and display associated significance levels with contours. The resulting SST pattern will then show the “real” interannual SST signal associated with interannual variability in Pf cases for the whole time period (El Nino should still show up with perhaps other interesting patterns in the other oceans). A similar analysis could be conducted using the low frequency components of the signals only, to show the decadal SST signal associated with the low frequency (decadal signal) in Pf cases. This linear analysis can then be complemented with composites (or anomaly patterns for discrete events eg 1997-98 etc) if the authors focus on singular Pf outbreaks.

Supp Fig 8 and Supp Fig 9: The authors use the CAMS GCM in AMIP mode (atmosphere only driven by prescribed SST conditions for which the trend has been removed) to look at associations between surface temperatures (land and ocean) and ENSO or PDO modes (before and after the slowdown). Why? A similar analysis could be conducted using temperature observations (using CAMS, CRU products etc) instead (by removing the linear trend first) or climate reanalysis (NCEP, ERA5...)? GCM in AMIP mode are ok, but they still suffer from significant biases, while there are several gridded observed dataset that could be employed to look at this properly. On top of that – the external forcing stops in 2005 and is replaced by the RCP6 scenario after that. I understand the authors wanted to use this simulation as it implicitly removes the long-term climate change signal – but that section is quite suspicious (results only rely on one GCM, no ensemble – the El Nino temperature response in the Tropics tend to be relatively uniform if you use observations etc etc).

Minor comments

L20: “dominant control of intervention measures.”

L23: "for both Plasmodium falciparum and vivax" – mention "Plasmodium" when first mentioned in the text – then use "P. falciparum" and "P. vivax" standard in the text – and please do italicize all biological species names in the text (like P. falciparum or An. Gambiae etc) as well. It looks ok later in the text (L81-82).

L44: "...within these vectors 1-8". Unsure it fits here – but when you mention the relationship between the sporogonic cycle and temperature you should cite the recent relevant work by the Penn state team (Waite et al. 2019 and Shapiro et al., 2017). They showed lower temp thresholds than previously estimated for the development of P. falciparum in key Anopheles vectors in a lab setting.

L54: "...the results of aggressive public health intervention..." there might be a better way to write this, I am unsure about "aggressive" perhaps "the results of extensive public health intervention".

L58: "...in driving reduced epidemics in the background...". Perhaps mention that climate plays a role on malaria transmission in the background, but control efforts make a big difference (antagonistically or synergistically depending) early in Intro for clarity.

L74-75: "with dynamical contributions of the major oceanic drivers of climate variability, such as ENSO and the PDO" perhaps precise "major oceanic drivers of natural climate variability" to underline the distinction between anthropogenic effects vs natural modes like ENSO and the PDO.

L81: "of extensive retrospective records on malaria cases" perhaps mention the time period for the available clinical data – to strengthen your statement

Fig 1 caption: minor comments: you use Fig 1a (minor case) then Fig 1A (major case for legends and Fig 1) so something to harmonize later. You can also update the caption: "Plasmodium vivax (Pv, blue line) and Plasmodium falciparum (Pf, green line)".

L107: "are followed by an overall decrease in cases" then "on these general trends (Fig. 1b)."

L111-112: remove opening and closing brackets – not needed I think

L118: "Methods and 7" – superscript 7 as I assume you refer to reference 7

L121: remove "As their names indicate" – it is quite implicit.

L141-L144: "The former... basis of temperature". Ok but I think the authors could repeat info there, as I found this sentence very confusing "The training set was used... the prediction set was used..."

L153: "as an explicit driver of the transmission rate" – ok but perhaps "mean monthly temperature is used as an explicit, standalone input of the transmission rate model"

L160: "significant and stronger for the second peak which is also the main malaria season". General comment – be very specific when you either describe seasonality in climate variables or in malaria cases.

L166-167: "can be explained solely on the basis of changes in the forcing by temperature of transmission dynamics." Ok but I think you can just say "can be explained solely on the basis of changes in the temperature forcing." I think you tend to use very long sentences – better to break them down for clarity when applicable/do-able, in particular for the sections highlighted in yellow.

L167: "start diverging in 2004..."

L171-172: "about malaria infections" – be more precise here (specify Plasmodium parasite) because incubation periods in human hosts for P. vivax and P. falciparum can differ. Plus there is a 9 months relapsing issue with P. vivax...

Adjust time tick marks on Figure 2 (either remove tick marks on Fig 2c or use same style than Fig 2a) and Figure 4 (Fig 4c looks messy – e.g. adjust "C" and remove time tick marks on Fig 4c or use same style as Fig 4a please)

L196: "further exhibit a low-frequency maximum I May-June 1997" Unsure about what you meant here, typos and sentence to double check – please re-word/clarify

L197-201: This is quite wordy (and might be related to other reviewer's comments) – same comment applies to the updated paragraphs highlighted in yellow. One suggestion to package the same paragraph/info in one sentence:

"The impact of the large El Niño events in 1997/8, 2002 and 2006 on both regional climate and malaria cases in DZ is shown at seasonal and interannual time scales (Fig. 2b,c and 4a,b)."

Avoid lengthy statements such as "We have overlaid on these panels the time windows corresponding

to the El Niños of 1997/98, 2002 and 2006 blabla” as this information can be briefly mentioned in figure caption or in method section.

L268 vs L271-272: “Similarly, the hypothesis that changing reporting rates may explain the turn-around is highly unlikely” vs “Increases in the number of reporting clinics are another possible confounding factor”.

? If you increase the number of reporting clinics – this will obviously impact the amount of surveillance in the region – so it’s a contradicting statement – please clarify

References

Shapiro LLM, Whitehead SA, Thomas MB (2017) Quantifying the effects of temperature on mosquito and parasite traits that determine the transmission potential of human malaria. PLoS Biol 15(10): e2003489.

Waite Jessica L., Suh Eunho, Lynch Penelope A. and Thomas Matthew B. 2019 Exploring the lower thermal limits for development of the human malaria parasite, Plasmodium falciparum Biol. Lett. 1520190275

Reviewer #5:

Remarks to the Author:

NCOMMS-19-37665A: Malaria trends in Ethiopian highlands track the temporary ‘slowdown’ in global warming at the turn of the century

The paper addresses an important topic and adds to the long-standing debate about climate and malaria in African highlands.

I find the paper a bit hard to read, and it could benefit from some reorganization and adding a few references. For example, line 39 refers to “other epidemic regions” but this could be misleading. We can have areas with stable malaria transmission called an epidemic as well. But these are not the areas this paper refers to. I missed the term “unstable malaria”, which only appears on the Results, line 102. That distinction is critical so the reader knows the type of malaria the manuscript refers to.

The same on line 78. Is the decline in malaria in Ethiopia observed across the country or only in the highlands?

Also, line 83, I would refer to widespread public health interventions, not intense, and I wonder if this refers to malaria control interventions or much broader public health interventions that could contribute to malaria reductions. Important to clarify. Also, adding a reference that supports the statement that there we limited interventions before 2005 is important.

Line 95 – the highlands mentioned here are the one in Ethiopia or is this a generalization comment?

Lines 165/6 state that “...the slowdown in reported cases at the beginning of the decade can be explained solely on the basis...” I think solely is too strong of a statement. Interventions could be limited but were present, and could potentially have a better effect because climatic conditions were favorable. But the results do not allow one to say that no other factors played a role. Besides, there is a lot of uncertainty in the results (any model has uncertainty), and making such a statement is a bit of a stretch. In fact, later in the manuscript, lines 255/6 make a much more reasonable statement, better aligned with the results presented in the manuscript.

Similarly, the statement on lines 260/1 is too strong. Malaria transmission is a complex process, and even if climate is the most important driver at some point, it is certainly not the only one. Migration,

for example, was not mentioned at all. But there are many issues. The discussion provided in Caminade et al (www.pnas.org/cgi/doi/10.1073/pnas.1302089111) makes an extremely important discussion on exactly this issue. This paper is not cited in the manuscript.

A policy change that introduced malaria control intervention is mentioned on line 281. But when was that change? Which year? Coverage? Line 349 mentions 2005 – are those two sentences talking about the same thing?

The paper focuses on *P.falciparum*, but results are discussed considering both *P.falciparum* and *P.vivax*, although on lines 289 and 290 it is suggested that a similar modeling work could be done for *P.vivax*. So, to what extent one-parasite model is valid for both? And if is, why model them separately. This is a bit confusing in the paper.

Lines 335/7 – “These conclusions are especially relevant under the current targets of malaria elimination, the successes mainly in epidemic, low transmission, regions, and the increase of *P. vivax* infections, a neglected public health problem.” Which successes are the authors talking about? When did they happen? These successes and the referred increase of *P.vivax* are in Ethiopia? When did the increase happen? Reference?

Line 422 – the issue of asymptomatic infections needs more discussion. Acquired immunity in an unstable malaria area is extremely weak. That is why it is unstable and prone to epidemics. How important is that in the Ethiopian highlands? References to support its importance?

Figures - What is the blue shaded area in Figure 2?

Lastly, I am wondering if one of the Ethiopians mentioned in the acknowledgement could be a co-author. This is something for the authors to think about, in light of the many decolonizing efforts going on in public health.

Minor comments:

Line 96 – intervention should be plural

Line 140 – replace than by as

Line 152 – replace low by limited

Line 168 – replace string by widespread

Line 210 – replace thee by the

Line 223 – remove the comma after the word study

Line 251 – replace malaria’s by malaria

Line 314 – replace Sub Saharan by sub-Saharan

Line 330 – there is a word missing after been (made, I think)

Line 417 – remove the comma

Line 483 – *Anopheles* needs to be in italic font

Reviewer #6:

Remarks to the Author:

This paper by Rodo et al. aims to provide an evidence that malarial transmission in a location of Ethiopian highlands is linked to climate and the downward trend at the turn of the century is linked to the global warming slowdown via changes in ENSO and PDO. I am reviewing this in the capacity as an expert in climate not disease dynamics, and saw that the authors had put in effort to address the previous comments of other reviewers. Based on my evaluation, the paper still requires revision and

clarifications.

It is appreciated that the signal decomposition using the SSA method is intended to reveal the synchronies between the variables. However, to be convincing, the signal components are ought to be at least made visible in the raw time series of the local temperature shown in Fig. 1B which is argued to be the important variable that drives variability in the malaria cases. This need not require a complex statistical technique such as SSA which many general readers would not be familiar with and the result is hard to verify. It is not to say that the authors should not use SSA, but as a first approach the three components: seasonal, interannual, and decadal can be all shown in a figure using common filtering techniques, even a simple one like running mean. For example, an 11-yr running mean can be used to highlight the low-frequency variability and trends, then this plus the seasonal mean can be subtracted to highlight the interannual component. The SSA can be conducted to further reveal the synchronization. Otherwise just by looking at Figure 1B, there is no association between local temperature and malaria cases, and the readers would need to believe solely on one statistical technique (SSA) disregarding the possibility of statistical artefacts that can arise from such complex techniques (e.g., overfitting, frequency leakage).

The relationship between local temperature (and precipitation) and the large-scale climate variables (ENSO, PDO) needs to be clarified, and how it links back to the primary topic of global warming slowdown. As the authors are aware, the PDO and ENSO are not independent of each other. The interannual component of the PDO encapsulates ENSO. The PDO that is said to be linked to the global warming slowdown refers to its decadal component. The PDO signal shown in Figure 4 appears to be the interannual component which should certainly coincide with ENSO. From this, it is not clear how it links the global warming slowdown which is supposed to correspond to the decadal component of the PDO (e.g., see Figure 29a in Santoso et al. 2017 <https://agupubs.onlinelibrary.wiley.com/doi/full/10.1002/2017RG000560>). Then in Supplementary Fig. 8, 9 (correlation between ENSO or PDO with grid-point surface air temperature) the authors showed that the influence of ENSO (PDO) weakens (strengthens) after 2000. But taking such a short period (2000-2016) to infer correlation with the decadal component of the PDO is not valid (it is only half the period), and the increased correlation may just be picking up the unresolved near-term trend. Are these correlations statistically significant? Significance test needs to take into account the auto-correlation of the time series which affects the effective degree of freedom (Ebizusaki 1997 <https://journals.ametsoc.org/jcli/article/10/9/2147/28747/A-Method-to-Estimate-the-Statistical-Significance>). Have the seasonal means and trends been removed prior to the correlation? Even the ENSO index contains trends which can affect the correlation in each of the periods. After doing all these, Supp. Fig. 9 and 8 would look similar.

How about performing correlation between ENSO index with the local temperature time series before and after 2000, and between the local temperature time series (and ENSO index) vs Pv, Pf?

What is rather puzzling is how to connect the trend in local temperature with the recent negative PDO (surface cooling trend over tropical Pacific) when the Indian Ocean has been warming up persistently (e.g., Roxy et al. 2014 <https://journals.ametsoc.org/jcli/article/27/22/8501/34834/The-Curious-Case-of-Indian-Ocean-Warming>). Figure 2d of England et al. (2014; already cited in the text) clearly shows warming over Africa, but cooling of the tropical Pacific. Please clarify and reconcile.

The argument that El Nino effect is decreasing after 2000 worth more attention. Apart from weaker El Nino events (e.g., more Central Pacific El Nino as Referee 4 mentioned), the period post 2000 has had more La Nina than El Nino (e.g., Figure 3 in Santoso et al 2017), and that could contribute to more local warming and thus less outbreaks. Reviewer 4 provided references on the change of ENSO which should be considered.

In Summary, the present manuscript has not clearly linked the downward trend in malaria case to the global warming slowdown.

Other comments:

L33: "output of numerical simulations of a global climate model" – note that CAM is an atmospheric model, not a full climate model. An atmospheric model has no full interactions with the ocean, and the sea surface temperatures are prescribed rather than freely evolving. The authors 'examined the consensus of atmospheric model simulations' (L236), how many ensemble members are examined?

Which temperature variable goes into the dynamical disease model? The one in Fig. 1B or the reconstructed components?

L186-190: Need to elaborate here what ENSO and PDO actually do to climate in the region of study.

L89 would read better like this: "In particular, a process-based transmission model is used to predict what would have been the effect of the observed change in temperatures on falciparum malaria cases in the absence of the public health interventions introduced post-2004."

L69-76: "The warming slowdown... in addition to a result of statistical discrepancies from different data sets and definitions of the phenomenon... Irrespective of the final causes behind it, the documented global took place.." - this part needs a reword. Statistical discrepancies and definitions cannot physically cause the warming slowdown that actually took place.

L121: "real trend" – not clear

L232 and elsewhere: "HadCRTU4" -> "HadCRUT4"

Supp. Figure 8-9: "Spatial correlations" – the analysis does not show a spatial correlation. It shows temporal correlation between the index and surface air temperature at each grid-point on the map.

L249: "Decadal variability in malaria cases for both parasites, P. falciparum and P. vivax, tracks the trends". – Not apparent in Figure 1

Responses to Referees

(Referees' comments in bold).

Reviewer #1 (Remarks to the Author):

I think the paper improved since the last round of revisions. My overall impression is that recent updates (related to former round of reviews) sometimes support the manuscript further, sometimes it can be quite the opposite. There are still points to clarify further before the manuscript can be accepted for Nature Communication, thus major revisions. See my comments in the following for further details.

We thank the referee for the positive comments and for finding the manuscript improved by the extensive revisions. We provide now additional clarifications and analyses. In particular, we clarify for the referee methodological considerations which are required to understand Fig. 5. We also show that actual correlation values and their significance support the conclusions on the reconstructed components and their association between malaria and climate variables.

Major comments

L124-127: paragraph commenting the match between climate variable and Pf/Pv cases at different time scales (seasonal vs interannual vs decadal). The authors mention a good match at all periodicities between climate variables and Pf/Pv cases – this statement is backed up by eyeballing time series on Figure 2. I think the authors should provide a more robust metric to assess this “match” further: simple correlations between climate and case RCs at all frequencies with associated significance levels could be provided somewhere in the manuscript (text and/or table in Supp Mat) to strengthen this point I think.

Thank you for the suggestion. We now provide the statistical descriptors requested by the reviewer in Supplementary Table I (below) and refer the reader to these values in the caption of the Figures. As the referee can see below, correlation coefficients and associated p-levels further support the evidence of a strong association for all the comparisons we had presented previously (for malaria incidence and climate indicators). In particular, we note the high correlation values, all significant with $p < 0.01$.

Supplementary Table I. Correlation coefficients corresponding to the time series for the reconstructed components shown in Figure 2. Significance levels are given for a randomization test which accounts for the decrease in the d.o.f. The lag corresponds to that for the maximum correlation is attained. We also show the corresponding panel of Figure 3 with the variables for which the correlations values were calculated. LT and ann refer to long-term and seasonal components, respectively.

Variables	Correlation coef.	p-value	Lag	Figure in manuscript
T_{min}, P_v (LT)	0.986	<0.01	0	3A
T_{min}, P_f (LT)	0.978	<0.01	0	3A
P_f, P_v (LT)	0.981	<0.01	0	3A
T_{min}, R (ann)	0.724	<0.01	1	3B
P_f, P_v (ann)	0.956	<0.01	0	3C

Figure 5: I think there is a serious methodological issue here. The authors correlate global SST data with SDC outputs for Pf falciparum cases over a 24 months window (in 1997-98 so during one of the largest El Nino on record). Thus, the correlation pattern obviously maximises an El Nino pattern (by construction! – a composite analysis might yield similar results as well). Importantly, the analysis is not conducted for the whole duration of the time series. If I had to do this properly, I will extract the interannual component of the Pf case data (already done here). Then I will filter out the low frequency component in the SST signal (to only retain the interannual variability) and then I will correlate them over the whole time period using lags and display associated significance levels with contours. The resulting SST pattern will then show the “real” interannual SST signal associated with interannual variability in Pf cases for the whole time period (El Nino should still show up with perhaps other interesting patterns in the other oceans). A similar analysis could be conducted using the low frequency components of the signals only, to show the decadal SST signal associated with the low frequency (decadal signal) in Pf cases. This linear analysis can then be complemented with composites (or anomaly patterns for discrete events eg 1997-98 etc) if the authors focus on singular Pf outbreaks.

We must respectfully disagree with the referee on this comment which follows from a fundamental misinterpretation of how the method (spatial SDC) works and what it was developed for. One of us (X. Rodó) was the developer of this method for the analysis of local correlations in time, and also in time and space. We refer the referee to Rodó, 2001; Rodríguez-Arias and Rodó, 2004, Rodó and Rodríguez-Arias, 2006, as well as to the following papers for applications in epidemiology (Rodo and Pascual, PNAS 2005; Rodó et al., PNAS 2014; Remais et al., 2009; Carroll et al., GCB 2016).

Importantly, the method seeks to identify associations (linear correlations in the application here) that are transient (that is, localized in time). But before we explain why some of the concerns and suggestions by the referee do not apply, we note that we do not correlate global SST data with SDC outputs for Pf cases. Instead, we apply SDC and obtain temporal correlations between the time series of SST at a given grid point of the global ocean and the time series of Pf

cases, for a given window of time. The method was developed to specifically determine whether a **local** observed correlation could arise at random, or whether it reflects instead a transient association (for that specific time period and the specific grid point) that is significant at a given level. It is important to note that the method has been extensively described, tested, and demonstrated. Its purpose is to detect local associations in time and not global ones (averaged over all the time of the time series or spatio-temporal data), in the same sense that wavelet analysis is localized and not global like the Fourier spectrum.

We refer the reviewer to the above papers to better understand the method in its details. Here we start with a published figure (below) to summarize the basics of how the method works when analyzing local correlations between two time series (so that our arguments below can be better followed, but also so that we underscore that the method does not generate spurious patterns but identifies the occurrence of existing frequencies that are common to the two data sets).

The way to look at the analysis is as follows (see plot on the left): (1) the two time series are displayed to the left and top of the matrix of correlation values (here as an example, the synthetic time series correspond respectively to the sum of periods 10 and 50 on the left, and to a sequential occurrence of period 10 only and then period 50 only, on top); (2) the matrix shows the positive correlations (in one color, gray) and the negative ones (in another, red), for a moving window (25 units for illustration) whose coordinates (x,y) in the matrix correspond to the respective location of such window along the time series on the left and top respectively; (3) optionally, the time series at the bottom shows the highest correlation for a given time (vertically) for the data set that acts as the driver; the dots on this time series indicate significant correlations at a given significance level). The way to “read” the results is to start from the diagonal, and to move vertically off from it, to identify a given lag corresponding to the first significant correlation that appears (off the diagonal).. We apologize for this brief summary here but we feel it is important that the referee follows the basics of this method which is “data-

adaptive”, in the sense that it does not require pre-processing of the data with filters etc. The published example above (from Rodríguez-Arias and Rodó, *Oecologia*, 2014) illustrates in particular how the analysis “picks” up the scale (frequency) of the association, only when the same frequencies are present, and therefore no predetermined shape is defined by default.

Let us now address more specific concerns of the referee and show that they do not apply, as already demonstrated in previous work. We start with fully temporal examples and follow with spatio-temporal ones.

- 1) We constructed (below) another synthetic example for the referee, to correlate a synthetic time series (e.g. of period 10, on the left) with the same ENSO data set we used in the paper (top), for a window size of 24 months. As seen, there are no correlations arising from the analysis; the few places where some value other than zero shows up are all non-significant (based on the randomization test applied). This is just an example; there are many more in Rodó’s publications on the method.

- 2) More to the point of interest here, in a second example we illustrate that our specific purpose is to identify local correlations, because the association between climate variability and infectious diseases can be transient and vary in its intensity for different climatic events (e.g. ENSO events). (This is also why others have used Wavelets in epidemiology and climate analyses, and shown this kind of transitory associations, see for example, Cazelles et al., 2005; 2007).

In particular, we consider here the Niño3.4 time series (top, time running to the right) and the raw time series of cases for *P. falciparum* (left, time running down). The moving window length is 43 months (as 3.6 years is the dominant frequency of the climate driver). The positive correlations are now in orange. We call the attention of the referee to particular times and events with the circles. Those in black indicate high (and highly significant correlations) for EN 86(-LN87) and EN97(-LN99); whereas the red circle indicates the lack of correlation for the 1991-95EN. In other words, one could average across time, but this would fail to note that the coupling between the two systems is strong and significant at some times but not others. Moreover, there is no predefined pattern that should arise ‘by construction’. We have added a version of this figure in the Supplement (Suppl. Fig 7), to indicate that the years of strong correlation correspond to those when Niño3.4 shows a strong effect also on the regional climate variable of interest (SST/temperatures) for Ethiopia.

Figure 5 in the manuscript takes the method into the spatial domain to repeat the local correlation analysis for the given time series (say malaria cases) and that of SST at each grid point of the global ocean. The significance is then determined for a given location and the map shows in red (blues) tones only those points where there is a significant positive (negative) correlation (with a binomial test plus a randomization test, see details in the methodological references). The coherence in space emerges from the analysis, and is indicative in itself of a meaningful result. (By coherence in space, we mean here a contiguous region and/or a region that

corresponds to a particular part of the ocean known to reflect, and be part of, a given phenomenon, such as El Niño)

- 3) To further clarify this point, we show below the differential spatial signature of each EN event from 1968 to 2006, as reflected in the global grid of SST anomalies.

(We apologize in advance if the size of the plots makes it difficult to read the details. The relevant patterns do show, and the corresponding figure in the Supplement is provided in a bigger format). This application of SDC to spatio-temporal data shows the maps of significant correlations between the Nino3.4 time series and the SST grid of the global ocean (i.e. the time series at each point of the grid). There are different plots to show results as we place the window (of 43 months) at different El Niño events, ordered from the top left to the bottom right according to the power of that frequency (3.6 years) for the given event. Each panel refers to an individual EN, ordered from major to minor EN intensity. Therefore, the top left window corresponds to the 97-98EN (the zero lag is displayed for all EN peaks). The second panel in the top row corresponds to the 81-82EN, etc.. Finally, the centered bottom plot shows the average “composite” of the correlations for all the events.

Without going into details, what is clearly apparent and we would like to convey to the referee is that the maps differ for the different events, and that this information would be lost if we averaged over time. See also that the spatial pattern of correlations and not only the values vary widely, with for example, a clear region of positive associations in the Indian Ocean close to

Ethiopia that is present in the two events we highlighted above, but is absent instead in many other events.

4) To finish with our argument, please consider below the SDC maps now calculated between the P_f cases in Ethiopia covered in our study and a global grid of air temperature (T_{2m}) rather than SST. We consider again a time window of 43 months (localized as before on the 97-98 peaks of the disease data). The different maps in the panels consider the same four different lags between the climate variable and the malaria cases than those applied in Fig. 5. The important point we would like to make here is that the resulting spatial pattern of correlation is very similar to that for SSTs in Figure 5, even though air temperatures have no “pre-determined” ENSO frequency. Air temperatures show this pattern because of the temporary ENSO SST forcing operating in 1997/98, which makes them exhibit the same frequency and spatial structure even though one cannot invoke a predetermined pattern in T_{2m} . Again, the analysis retrieves the structure from the common frequencies to both air temperatures and P_f malaria cases. Thus, the pattern we obtained in Figure 5 is not a spurious result of SST “containing” the ENSO frequency. The pattern indicates instead a coherence of frequencies in the temporal domain between air temperatures and malaria cases at that selected moment in time, when the largest El Niño of the 1990s occurred. The fact that these correlations “pick up” a meaningful and contiguous part of the Pacific Ocean (related to the physics of the El Niño phenomenon) is an indication of a dynamically significant result and not a spurious one by construction.

In short, we hope the referee better appreciates the reasons why we chose to apply this method and to provide Fig. 5 in our manuscript, and especially why we did not replace it by a composite, and then followed with some form of localized analysis that requires pre-processing and manipulation of the data. The resulting SST pattern in the figure represents already the “real” interannual SST signal associated with the interannual variability in P_f cases.

Because we appreciate that some of the concerns raised by the referee may be shared by those not familiar with the SDC method, we have now added the two figures above to the supplementary material, to further illustrate and clarify the approach, and strengthen our argument. We have also extended the text associated with figure 5 to make the analysis and the results clearer. The captions of the supplementary figures provide additional methodological information.

Supp Fig 8 and Supp Fig 9: The authors use the CAMS GCM in AMIP mode (atmosphere only driven by prescribed SST conditions for which the trend has been removed) to look at associations between surface temperatures (land and ocean) and ENSO or PDO modes (before and after the slowdown). Why?

The rationale of our approach was that, given the relatively short intervals of time to be compared, we could make use of one of the best atmospheric models available to properly simulate the effects of these two main climate phenomena for the region of interest. Essentially as of today, good station data coverage is far from optimal in Africa including in this region. As the referee states, 'use of GCM in AMIP mode are ok' and what we sought here was to use solid GCM simulations where by removing the potentially confounding influence of trends, we could investigate the differential climate forcing over the region and provide further support and context for the analysis of observations (e.g. Figs. 1,3,4). The results are, as it can be seen, highly consistent with those of these other analyses, and the independence in the two approaches adds further strength to our findings. We have further clarified the purpose of these analyses in the text.

A similar analysis could be conducted using temperature observations (using CAMS, CRU products etc) instead (by removing the linear trend first) or climate reanalysis (NCEP, ERA5...)? GCM in AMIP mode are ok, but they still suffer from significant biases, while there are several gridded observed dataset that could be employed to look at this properly. On top of that – the external forcing stops in 2005 and is replaced by the RCP6 scenario after that. I understand the authors wanted to use this simulation as it implicitly removes the long-term climate change signal – but that section is quite suspicious (results only rely on one GCM, no ensemble – the El Nino temperature response in the Tropics tend to be relatively uniform if you use observations etc etc).

Thank you for the suggestion. However, as the referee knows well, climate records in Africa are sparse, often discontinuous with many gaps and inconsistencies. This is particularly the case for Ethiopia. Meteorological time series tend to be short and not properly curated, and therefore, reanalysis essentially recovers the physics of climate provided by the models employed to generate these variables. In a sense, this means that differences with our results for the observational periods should be extremely minimal. In our case, we preferred instead to use the more robust dynamical strength provided by atmospheric models, in an attempt to separate potential differences in forcing by ENSO (and the PDO, more so in this case given the slower pace of variability). In addition, as another referee had indicated in the former round of reviews and we stress above, the limited observational period (limited by the length of malaria records) constrains the scope of the analyses we can conduct (e.g. with the PDO as not even a full PDO cycle would be covered). It is in this context that we considered more robust to investigate changes in the forcing by different climate modes with the AGCMs in this part of the work. We

view the particular analysis as a valuable addition and a complement to the time series analyses presented earlier on in the manuscript.

In addition, both CAM 4.0 and 5.0 incorporate a number of enhancements to the physics package (e.g. several adjustments to the deep convection algorithm), and collectively these improvements yield a significantly improved atmospheric modeling capability, even though there are always aspects of the models that are not ideal. We would like to stress the clear signatures appearing in the analyses of our simulation outputs, even if we concede that biases do exist, as stated by the referee. Biases in the data would be much more problematic than those present in the model simulations (e.g. consider for instance the paucity of data records for this location in the seventies and eighties, even though they are the basis for the re-analysis mentioned by the referee). The potential biases in our model simulations would be systematic, and therefore affect similarly all the temporal intervals. We cannot see how these biases would produce a stronger EN forcing in the earlier part of the record (with a sparser data coverage) than later on (EN weakening after the 2000s).

We have now better explained our motivation and acknowledged potential limitations of the simulations.

We hope that these explanations make clear the added value of considering model outputs as a complement for the rest of the paper. We also hope that the referee is now satisfied with this part, and with the efforts made in the revision concerning other large-scale climate modes, including the role of the Indian Dipole Mode following his/her question in the previous revision. Thank you for raising these matters which allowed us to better explain our methods and rationale.

Minor comments

L20: “dominant control of intervention measures.”

Done. thank you.

L23: “for both *Plasmodium falciparum* and *vivax*” – mention “*Plasmodium*” when first mentioned in the text – then use “*P. falciparum*” and “*P. vivax*” standard in the text – and please do italicize all biological species names in the text (like *P. falciparum* or *An. Gambiae* etc) as well. It looks ok later in the text (L81-82).

There is some potential confusion here on the particular text in Line 23 of the abstract. Please note that we had written “*falciparum* and *vivax* malaria” because it is customary not to include the Genus (*Plasmodium*) when referring to the disease itself, malaria, due to a given parasite. That is, *falciparum* malaria stands for malaria caused by *Plasmodium falciparum*. In this kind of notation, it is also the case that the italic is not used as one is not including the species name as such. We had chosen this simpler writing in the abstract to make the sentence more readable. In the main text, we had used italic when using the species name and we had also defined “*P_v*” and “*P_f*” to simplify the writing, as for other variables. We have now changed the writing in the abstract to refer to the parasite species, and therefore in italic, and avoid this kind of misunderstanding. If the editors or the referee think that what we had was more appropriate for the abstract’s writing, we would be happy to revert to it. We have also checked that throughout the text we have used italic where needed.

L44: “...within these vectors 1-8”. Unsure it fits here – but when you mention the relationship between the sporogonic cycle and temperature you should cite the recent relevant work by

the Penn state team (Waite et al. 2019 and Shapiro et al., 2017). They showed lower temp thresholds than previously estimated for the development of P. falciparum in key Anopheles vectors in a lab setting.

Thank you for the references. We added them later in the text. They do not belong here at the beginning of the Introduction since the references we give have to do with the debate and findings on the trend in incidence.

L54: “...the results of aggressive public health intervention...” there might be a better way to write this, I am unsure about “aggressive” perhaps “the results of extensive public health intervention”.

Done.

L58: “...in driving reduced epidemics in the background...”. Perhaps mention that climate plays a role on malaria transmission in the background, but control efforts make a big difference (antagonistically or synergistically depending) early in Intro for clarity.

Following the suggestion of the referee we have changed the writing and added: “in the background of existing control efforts and before enhanced ones.” It is important to make clear here that we are looking at the effect of the slowdown in warming before the strong control efforts that started in 2004-2005. Of course, those enhanced or extensive control efforts that came later acted together with the changing climate conditions. But the sentence here in the introduction refers to the turnaround in 2000. We make clear in the rest of the paper and especially in the Discussion that control is important and eventually drives the cases to very low levels. Our point is that this effect of enhanced control acted in the background of climate conditions less favorable to transmission than those of the 1990s.

L74-75: “with dynamical contributions of the major oceanic drivers of climate variability, such as ENSO and the PDO” perhaps precise “major oceanic drivers of natural climate variability” to underline the distinction between anthropogenic effects vs natural modes like ENSO and the PDO.

Agreed, we modified accordingly.

L81: “of extensive retrospective records on malaria cases” perhaps mention the time period for the available clinical data – to strengthen your statement

Done. Thank you.

Fig 1 caption: minor comments: you use Fig 1a (minor case) then Fig 1A (major case for legends and Fig 1) so something to harmonize later. You can also update the caption: “Plasmodium vivax (Pv, blue line) and Plasmodium falciparum (Pf, green line)”.

Agreed, we modified accordingly.

L107: “are followed by an overall decrease in cases” then “on these general trends (Fig. 1b).”

Done.

L111-112: remove opening and closing brackets – not needed I think

Done.

L118: “Methods and 7” – superscript 7 as I assume you refer to reference 7

Yes, we modified the text to make this clearer.

L121: remove “As their names indicate” – it is quite implicit.

Agreed, we modified the text accordingly.

L141-L144: “The former... basis of temperature”. Ok but I think the authors could repeat info there, as I found this sentence very confusing “The training set was used... the prediction set was used...”

Done. Yes, this was unclear.

L153: “as an explicit driver of the transmission rate” – ok but perhaps “mean monthly temperature is used as an explicit, standalone input of the transmission rate model”

We adopted this suggestion which is clearer than our original text but removed “model” at the end since the sentence is about the transmission rate as a parameter of the model, and we start the sentence with “In the model,”

L160: “significant and stronger for the second peak which is also the main malaria season”. General comment – be very specific when you either describe seasonality in climate variables or in malaria cases.

Yes, this is important.

L166-167: “can be explained solely on the basis of changes in the forcing by temperature of transmission dynamics.” Ok but I think you can just say “can be explained solely on the basis of changes in the temperature forcing.” I think you tend to use very long sentences – better to break them down for clarity when applicable/do-able, in particular for the sections highlighted in yellow.

We agree. Good suggestion. Some extra wording was added in response by other, earlier, reviews, but this writing you suggest works much better.

L167: “start diverging in 2004...”

Done.

L171-172: “about malaria infections” – be more precise here (specify Plasmodium parasite) because incubation periods in human hosts for P. vivax and P. falciparum can differ. Plus there is a 9 months relapsing issue with P. vivax...

Right, we were referring to *Plasmodium falciparum*, the species for which the model was developed.

Adjust time tick marks on Figure 2 (either remove tick marks on Fig 2c or use same style than Fig 2a) and Figure 4 (Fig 4c looks messy – e.g. adjust “C” and remove time tick marks on Fig 4c or use same style as Fig 4a please)

Done accordingly.

L196: “further exhibit a low-frequency maximum 1 May-June 1997” Unsure about what you meant here, typos and sentence to double check – please re-word/clarify

Thank you, this sentence was removed to avoid confusion.

L197-201: This is quite wordy (and might be related to other reviewer's comments) – same comment applies to the updated paragraphs highlighted in yellow. One suggestion to package the same paragraph/info in one sentence:

“The impact of the large El Niño events in 1997/8, 2002 and 2006 on both regional climate and malaria cases in DZ is shown at seasonal and interannual time scales (Fig. 3B,C and 4A,B).”

We have reduced the text and adopted the suggested sentence which is indeed clearer

Avoid lengthy statements such as “We have overlaid on these panels the time windows corresponding to the El Niños of 1997/98, 2002 and 2006 blabla” as this information can be briefly mentioned in figure caption or in method section.

Agreed, we deleted the sentence.

L268 vs L271-272: “Similarly, the hypothesis that changing reporting rates may explain the turn-around is highly unlikely” vs “Increases in the number of reporting clinics are another possible confounding factor”.

? If you increase the number of reporting clinics – this will obviously impact the amount of surveillance in the region – so it's a contradicting statement – please clarify

We agree that this was somewhat confusing as written. The first sentence referred to increases in reporting efforts by the same clinics and not to the addition of reporting clinics. To simplify and clarify this text, we opted for removing the first statement and leave only the second.

References

Shapiro LLM, Whitehead SA, Thomas MB (2017) Quantifying the effects of temperature on mosquito and parasite traits that determine the transmission potential of human malaria. PLoS Biol 15(10): e2003489.

Waite Jessica L., Suh Eunho, Lynch Penelope A. and Thomas Matthew B. 2019 Exploring the lower thermal limits for development of the human malaria parasite, Plasmodium falciparum Biol. Lett. 1520190275

Thank you, we added these references

Reviewer #5 (Remarks to the Author):

NCOMMS-19-37665A: Malaria trends in Ethiopian highlands track the temporary ‘slowdown’ in global warming at the turn of the century

The paper addresses an important topic and adds to the long-standing debate about climate and malaria in African highlands.

Thank you for the appreciation of the work and for the constructive comments.

I find the paper a bit hard to read, and it could benefit from some reorganization and adding a few references. For example, line 39 refers to “other epidemic regions” but this could be misleading. We can have areas with stable malaria transmission called an epidemic as well. But these are not the areas this paper refers to. I missed the term “unstable malaria”, which only appears on the Results, line 102. That distinction is critical so the reader knows the type of malaria the manuscript refers to.

We have now edited and extended this opening text in the Introduction so that what we meant by epidemic is clear. We use instead “unstable” malaria and make its meaning clear from the start. (We had originally avoided use of this the term early in the writing, because there is some disagreement among the malariologists we interact with on the best terminology and indeed some colleagues prefer seasonal epidemic to unstable). Thank you for pointing out the possible confusion here.

We have clarified the points raised below and hope that the changes make the manuscript easier to follow. With respect to some re-organization, we have now inverted the order of the dynamical model and time series analyses. We think this does make the logical flow easier to follow.

The order of the analyses/results we have now is as followed is: (1) Motivated by the general patterns of temperature and malaria data in Figure 1, the malaria modeling work addressing what the effect of temperature would predict for the decade of 2000; (2) The statistical analysis of the correspondence between patterns of variability at different temporal scales (decadal/trends, interannual variability, and seasonality) for malaria cases and climate (3) investigation of the connections between the regional and global climate variability, in the context of changes related to climate change.

The same on line 78. Is the decline in malaria in Ethiopia observed across the country or only in the highlands?’

We have edited this sentence to be specific about the trends in the highlands of Ethiopia, as this sentence refers to the focus of our work, and to trends that are pertinent to the highlands. That is, the decline in the 2000s that followed the rise in the 1980s and 1990s.

The enhanced intervention efforts in Ethiopia went beyond the highlands.

Also, line 83, I would refer to widespread public health interventions, not intense, and I wonder if this refers to malaria control interventions or much broader public health

interventions that could contribute to malaria reductions. Important to clarify. Also, adding a reference that supports the statement that there we limited interventions before 2005 is important.

We now used “widespread” and specified that the interventions were malaria ones. We also provided references.

Line 95 – the highlands mentioned here are the one in Ethiopia or is this a generalization comment?

We now specify that we referred to East African highlands.

Lines 165/6 state that “...the slowdown in reported cases at the beginning of the decade can be explained solely on the basis...” I think solely is too strong of a statement. Interventions could be limited but were present, and could potentially have a better effect because climatic conditions were favorable. But the results do now allow one to say that no other factors played a role. Besides, there is a lot of uncertainty in the results (any model has uncertainty), and making such a statement is a bit of a stretch. In fact, later in the manuscript, lines 255/6 make a much more reasonable statement, better aligned with the results presented in the manuscript.

We have rewritten this statement to remind the reader that the temperature effect occurs in the context of existing control efforts. The text now reads:

“Thus, changes in temperature forcing in the context of existing, low, levels of control have influenced the slowdown in reported cases at the beginning of the decade.”

We agree that this statement differed from that in lines 255/6, but please note that there the emphasis was the synergy between temperature forcing and the enhanced control that came later (2005) years after the turnaround in cases that are the focus of lines 165/6.

Similarly, the statement on lines 260/1 is too strong. Malaria transmission is a complex process, and even if climate is the most important driver at some point, it is certainly not the only one. Migration, for example, was not mentioned at all. But there are many issues. The discussion provided in Caminade et al (www.pnas.org/cgi/doi/10.1073/pnas.1302089111) makes an extremely important discussion on exactly this issue. This paper is not cited in the manuscript.

We have rewritten those lines and added to the text, including the reference to Caminade et al: “Thus, the results of our transmission model indicate that changes in temperature could have driven, fully or in part, the reversal from an increasing to a decreasing trend in seasonal epidemic size. Malaria dynamics and its response to climate change are of course complex phenomena involving many other drivers, including land-use, migration, and socio-economic conditions (Caminade et al. 2014). As designed, our modeling analysis isolates the effect of temperature, all else remaining constant.”

We think it is important to recognize that our modeling analysis seeks to specifically consider the effect of temperature given that all other factors remain constant, to ask whether this specific factor can account for the observed change. We found that the typical predictions of our stochastic model match quite well the observations, albeit with uncertainty. The uncertainty ranges from a weaker effect than the one observed (which still exhibits a reversal of the trend) to a stronger one. We would be happy to further elaborate on this range if the referee would

like to see further discussion. When we wrote that temperatures on their own could explain the observed reversal of the trend, we meant it exactly in the sense that a model in which only temperatures change, is sufficient to predict the observed pattern (as its ‘typical’ outcome). We did not mean that malaria dynamics do not involve a complex set of influences. We hope this is clearer now. As Caminade et al. indicate in their Discussion, there is a need for regional/local studies that “validate” the findings of malaria projections under changing climate. This is exactly what we have done here by considering a training and prediction set at local scales, in a way that complements global modeling and scenarios whose comparison to data are of a very different nature.

A policy change that introduced malaria control intervention is mentioned on line 281. But when was that change? Which year? Coverage? Line 349 mentions 2005 – are those two sentences talking about the same thing?

Yes, they are. We have now added the year to the writing in line 281 to clarify this. Thank you for noticing that this was unclear.

The paper focuses on P.falciparum, but results are discussed considering both P.falciparum and P.vivax, although on lines 289 and 290 it is suggested that a similar modeling work could be done for P.vivax. So, to what extent one-parasite model is valid for both? And if is, why model them separately. This is a bit confusing in the paper.

We have now rewritten this sentence to emphasize that the results of the time series analyses are similar for both parasites, but that the transmission model was developed only for Plasmodium falciparum infections. We do explain in this text that the difficulty with a P. vivax model arises from the relapse that characterizes this parasite. We give a reference to a model of vivax malaria which could be considered in a future extension of this work. We have gone through the text to be sure it is clear on which analyses were conducted for both parasites, and which one was not. We hope this is now clear, thank you.

Lines 335/7 – “These conclusions are especially relevant under the current targets of malaria elimination, the successes mainly in epidemic, low transmission, regions, and the increase of P. vivax infections, a neglected public health problem.” Which successes are the authors talking about? When did they happen? These successes and the referred increase of P.vivax are in Ethiopia? When did the increase happen? Reference?

We have now added two relevant references that describe the progress in Ethiopia at the country level, and also a third one that analyzes reported cases of both parasites in unstable malaria regions of arid districts of India. This closing sentence refers to the implications of climate changes interacting with control, and also to the importance of considering climate patterns as a context for control, in regions where climate factors are strong drivers (“fringe” regions of unstable malaria transmission). These are also the regions where control efforts have been most successful. We have now specified the period of time, have been more specific about location, and have added some references. We have edited slightly the writing to mention the increasing realization of P. vivax as a neglected public health problem, to avoid generalizing about increases in incidence. Those clearly apply to arid malaria in India (as seen for example in the reference we provided) but not necessarily across all fringe regions. The successes refer to the lower reported incidence especially of P. falciparum in unstable malaria regions.

To be more specific here to respond to the referee’s question on which successes we are referring to, please consider that the Federal Ministry of Health of Ethiopia distributed 18.2

million LLINs from 2006 to 2007, 13 million in 2010, and 42.4 million between 2014 and 2016 (Tafese et al., 2018). At the same time, the country launched and scaled up a health extension program by assigning workers trained in malaria case management to each kebele. These interventions accounted for a 50 percent reduction in hospital malaria morbidity and 60 percent reduction in mortality between 2006 and 2011 at the country level (PMI 2018; Aregawi et al. 2014). These references are in the manuscript where we describe the Ethiopian interventions.

Line 422 – the issue of asymptomatic infections needs more discussion. Acquired immunity in an unstable malaria area is extremely weak. That is why it is unstable and prone to epidemics. How important is that in the Ethiopian highlands? References to support its importance?

We have now clarified in the Methods that this part of the model was included for completeness, and that the model fit should determine the relevance of this part of the transmission system:

“Such individuals should not make an important contribution to the force of infection in unstable regions; we include it here for completeness and let the data determine their importance and associated parameters. Previous work has indicated that a model that includes them better fits the data than a traditional SIRS formulation, even when their contribution to the overall force of infection is small (e.g. 60).”

We provide references for the use of this kind of model and the inclusion of asymptomatics in the model for other unstable/ low epidemic regions. These models have been applied to fit time series data (see for example reference 60 to Laneri et al. PloS Comp. Biol. 2010). For another fringe region (in deserts), we show there that a model that includes asymptomatics performs better than one that adopts a traditional SIRS formulation. The reasons are discussed. Most importantly, we provide here discussion in the Supplement, in the caption of the table in Figure S3, on the parameters associated with this class:

“Most parameters exhibit reasonable estimates with well-defined CIs. The duration of class Q deserves discussion as it can become arbitrarily low. This is not surprising because the role of this class is to add a reservoir of transmission via an asymptomatic class. The rate at which infected individuals flow into this class is much lower than that at which they return to S. Thus, at any given time the number of individuals in Q is quite small and not an important component of the dynamics. The model fit is therefore unable to constrain the lower limit of the length that individuals spend in Q as this parameter has only a small effect on the temporal pattern of cases. This is consistent with an asymptomatic class not playing an important role in regions of unstable transmission.”

We therefore do not provide references to support the importance of this class. The fit itself indicates that this class plays a small part in the dynamics. Even if this is the case, we consider the general model a more realistic one than an SIRS formulation, and we have demonstrated in other studies for other locations that it fits unstable dynamics better than SIRS. We hope the model description is clearer now.

Figures - What is the blue shaded area in Figure 2?

The blue region depicts the slowdown interval, the focus of our investigation. Thank you for noticing that this was not described in the caption. We have now added the description in the figure caption.

Lastly, I am wondering if one of the Ethiopians mentioned in the acknowledgement could be a co-author. This is something for the authors to think about, in light of the many decolonizing efforts going on in public health.

We fully agree with the referee on being as inclusive as possible in public health studies that rely on data collected by those in other countries. We have had long-lasting collaborations in India and Bangladesh, and are now working in Ghana as well as in Brazil. We are aware and sensitive to this issue; please see that all our papers have local collaborators, most often multiple ones. In this case, we have included Amir Siraj with whom we had worked before, as related to the data collection and regional expertise on malaria. We note that two other researchers were previously included as co-authors in two earlier papers concerning this region of Ethiopia and this data set. In particular, these researchers were included in our first publication for this region in the journal *Science*, and in the second one, in the *Proceedings of the Royal Society London B*. We have carefully considered criteria for authorship and feel the names we had included here are those that the actual work can justify.

Minor comments:

Line 96 – intervention should be plural

Done

Line 140 – replace than by as

Done.

Line 152 – replace low by limited

Done

Line 168 – replace string by widespread

We have replaced string by enhanced

Line 210 – replace thee by the

Done

Line 223 – remove the comma after the word study

Done

Line 251 – replace malaria's by malaria

Done

Line 314 – replace Sub Saharan by sub-Saharan

Done

Line 330 – there is a word missing after been (made, I think)

Yes, thank you for noticing

Line 417 – remove the comma

Done

Line 483 – Anopheles needs to be in italic font

Done.

Reviewer #6 (Remarks to the Author):

This paper by Rodo et al. aims to provide an evidence that malarial transmission in a location of Ethiopian highlands is linked to climate and the downward trend at the turn of the century is linked to the global warming slowdown via changes in ENSO and PDO. I am reviewing this in the capacity as an expert in climate not disease dynamics, and saw that the authors had put in effort to address the previous comments of other reviewers. Based on my evaluation, the paper still requires revision and clarifications.

Thank you for your additional comments. We hope the responses help clarify our work. These include at the end of this document, the main points of our work so that the referee can consider its contribution in the context of the broader climate and malaria research, especially that of the long debates on highland malaria and temperature trends.

It is appreciated that the signal decomposition using the SSA method is intended to reveal the synchronies between the variables. However, to be convincing, the signal components are ought to be at least made visible in the raw time series of the local temperature shown in Fig. 1B which is argued to be the important variable that drives variability in the malaria cases. This need not require a complex statistical technique such as SSA which many general readers would not be familiar with and the result is hard to verify. It is not to say that the authors should not use SSA, but as a first approach the three components: seasonal, interannual, and decadal can be all shown in a figure using common filtering techniques, even a simple one like running mean. For example, an 11-yr running mean can be used to highlight the low-frequency variability and trends, then this plus the seasonal mean can be subtracted to highlight the interannual component. The SSA can be conducted to further reveal the synchronization.

We thank the referee for this remark which led us to incorporate the trend in T_{\min} in Fig. 1 (B,C). The reader will be able now to easily see how this low frequency component is related to the overall dynamics of the two malaria parasites infections (Fig. 1C; see below). For clarity, we did not add the interannual component here as the figure is very busy already.

We show below the new panels of Figure 5 including the trend. The red line indicates the trend and is exactly the same in the two panels, but with a different axis range. The different “look” of this same trend when visualized with different axes may have given the false visual impression to the referee that the trend in T_{\min} was not alike that of the malaria cases. This figure now clarifies that potential mis-impression and also underscores that small differences in temperature can result in apparently “large” changes in the scale of cases, given the nonlinear response of the epidemiological system to this climate factor, which is known from the

physiology of the malaria vector and that of the parasite. We hope this is much clearer now, thank you for the suggestion.

We also include below the requested separation of components with the moving average approach for your perusal.

As you can see, a 11-yr moving average provides a rough approximation to the low-frequency component (or trend) in T_{\min} (top-right panel). This component also appears as expected, very similar to the one obtained with the SSA decomposition, albeit much more noisy due to the fact that -conversely to SSA- there is no objective rule to decide that 11 years is an appropriate window, not 12 or 13. Also, the constraint of applying a window does not allow us to obtain the trend much beyond the turn of the century.

Although we appreciate the suggestion of applying a smoothing algorithm as a simpler analysis, we feel this would be effectively redundant and rely on a more idiosyncratic approach than SSA, which was specifically developed for the decomposition of variables arising in the dynamics of nonlinear systems. If after our response here the referee still feels there is something to gain from showing the above results, please let us know and we will add this figure to the SI.

We would also like to note that SSA should not be viewed as a particularly complex method of time series analysis. The method was established a long time ago and was built on the basis of early methodology (going as far back as Karhunen, 1946 and Loève, 1945, Takens 1981, and later Broomhead and King, 1986, Vautard et al., 1992, who expanded its applications to users from different fields, to offer just a few citations). Among non-parametric methods, SSA has proven particularly valuable for separating variability among frequency bands, and as such has been widely adopted in many fields now for about three decades. Beyond its use in climate research, SSA has also become a standard tool in the analysis of biomedical, mathematical, geometrical and several other time series (Golyandina et al., 2010; Hassani et al., 2015), with applications for example even in genetics (Sanei et al., 2011; Du et al., 2008; Tang et al., 2012).

In summary, the application of SSA is well-suited and rigorous for our purposes, and should be amenable to the readers at least to understand and follow the meaning of the outputs. We provide technical details on the methodology. The results from the spectral decomposition and the ulterior reconstruction are statistically very solid (further details and value statistics are now added in the figure captions).

Otherwise just by looking at Figure 1B, there is no association between local temperature and malaria cases, and the readers would need to believe solely on one statistical technique (SSA) disregarding the possibility of statistical artefacts that can arise from such complex techniques (e.g., overfitting, frequency leakage).

Our modification of Figure 1 should now address this issue, as it clearly illustrates the visual association between the variables. Also, as pointed out above most analyses on climate and infectious disease do not typically expect simple, linear patterns of association, as “small” changes in temperature can lead to changes in incidence of different magnitude depending on where in the range of temperatures one is located. Visual inspection at best reveals that there is a trend in the malaria cases, and also in the temperature data (although here the importance of seasonality may make this pattern less apparent). Our approach had been to decompose the time series first, in what we consider an exploration of potential coherence and at different time scales. On this basis, we then constructed a process-based model that incorporates the temperature as the driver of transmission, exactly to ask whether we could explain the patterns in disease with those in temperature. Nevertheless, we found the suggestion of the referee a real improvement of Figure 1 which makes the potential association clearer and better motivates the work that follows. We also now changed the order of the major results in the text, to show first those on the transmission model and follow then with those of the SSA decomposition and associations at different temporal scales. We hope this inversion of the order helps with the clarity of the presentation.

We respectfully disagree however on the suggested artifacts as a concern for the patterns of association we have presented. Following the suggestion by referee 1, we have added the correlation values and their significance for the different reconstructed components of Figure 3 in a new Table. These values support the similarity illustrated by visual inspection of the components in the figure.

In particular, we do not see how overfitting would apply in this case, given that the resulting SSA Reconstructed Components have not been used for prediction purposes. Please note that SSA has been applied in several disciplines to get rid of the overfitting problems produced by other prediction methods. As already mentioned though, the RCs extracted here both from environmental and malaria time series were presented in our manuscript as an exploration of association at different time scales; they were not used for prediction, which was instead addressed with the compartmental malaria model. Also, our malaria model did not use the output of SSA but the temperature data without any prior filtering or decomposition.

Moreover, the particular use of SSA was motivated in our work by the considerable and well-defined portion of variability accounted for by the seasonal, interannual and decadal components shown in Figures 3 and 4. Further motivation from exploring the analysis was provided by the clear evidence of underlying nonlinear dynamics, as seen in the presence of pairs of eigenvalues that correspond to periodic components of a clear frequency. To further

strengthen the presentation of the results, we now include text on the significance values of the RC used in our study (based on both white and red-noise null models).

Regarding a potential ‘leakage’ problem, there is no reason to expect this to happen. It is impossible for this analysis to generate a non-systematic bias that could produce exactly the ‘same’ dynamical components in different and totally unrelated series at different scales. More specifically, the spectral tools employed after the SSA decomposition, such as MTM, attempt to substantially reduce the variance of spectral estimates, by using a small set of tapers (Thomson, 1982, Percival and Walden, 1993) rather than the unique data taper or spectral window used by Blackman-Tukey methods. In practice, only the first $2p - 1$ tapers provide little spectral-leakage (Slepian, 1978; Thomson, 1982; Park et al, 1987), and the recovered seasonal, interannual and decadal components are contained within this range. It is also well-known that longer datasets -and ours would already lie within this category- can admit the use of a greater number of tapers (K) while maintaining a desired frequency resolution, and the optimal choice of K and p is in general application specific.

Nevertheless, to consider the referee’s concern on this matter, we compared our spectral results with those of an even more “leakage-resistant” spectral estimate (e.g. the adaptively weighted multitaper spectrum). Here, a weighting function further guards against broadband leakage for a colored but locally-white process. Results did not show any change, as expected from the lack of locally-white noise segments in our time series. We therefore retained our former results, as they offer a good compromise between the required frequency resolution to resolve distinct climate signals (e.g., ENSO and decadal-scale variability) and the benefit of multiple spectral degrees of freedom (see e.g., Mann and Park, 1993).

Finally, we agree with the referee that we would not expect the readers to “believe” our conclusions on the basis of a single analysis, and we hope that the text does not give the impression that this is the case, as our argument is further from resting uniquely or largely on the SSA results. We have presented coherent results obtained by means of two different (and methodologically independent) time series approaches, namely SSA in the frequency domain and SDC in the time domain. A third independent approach, the dynamical malaria model, provides further consistent evidence on a more mechanistic basis.

The relationship between local temperature (and precipitation) and the large-scale climate variables (ENSO, PDO) needs to be clarified, and how it links back to the primary topic of global warming slowdown.

We thank the referee for this important point. We realize that we had not provided sufficient context from the literature on the linkages between regional climate in East Africa/the Indian Ocean and large-scale climate processes. We have now expanded on this topic some in the Introduction but mainly in the Discussion. We refer the referee to the new paragraphs of text in the Discussion, and also to the recent Science review we now cite. (Cai et al. 2019) (and the commentary in the same issue), which provides an accessible context for general readers in a way we could not hope to do here. We have further emphasized our motivation for focusing on ENSO and its changing influence over the region of interest. This motivation should now be clearer; it follows from the central place of the tropical eastern Pacific (and therefore ENSO) on the redistribution of heat related to the slowdown in global warming. Interestingly, this redistribution involves complex and bi-directional feedbacks between the different tropical ocean basins and between decadal and interannual El Niño time scales, in ways that are

increasingly recognized but not completely understood. Changes in ENSO and in the frequency and intensity of EN and LN are part of it, in a consistent way with our findings, including the varying nature of EN event and their influence on the Indian Ocean and back on the Pacific Ocean (and for us, malaria).

With regard to the existing literature on linkages between Pacific SSTs and Ethiopian climate, we point out that the focus has been on precipitation for the country or the Great Horn of Africa. That is, numerous studies have shown significant correlations between El-Niño years and a deficit in rainfall locally (Beltrando and Camberlin 1993; Diro et al. 2010; Gissila et al. 2004; Hansen et al. 2011; Korecha and Barnston 2007). Other studies investigated the relationship between the Short Rains and the El Niño- Southern Oscillation (ENSO; Ropelewski and Halpert, 1987; Ogallo, 1988; Hastenrath et al., 1993; Mutai et al., 1998; Nicholson, 2017). The mechanistic links are much less understood (but see Diro et al. , 2011). There have only been a few other studies conducted on the relationship between decadal rainfall variability in Ethiopia and global SSTs. Motivated by the increased frequency of drought in the last decade, some studies emphasized the recent downward trend of the Long Rains over parts of the Great Horn of Africa, GHA (Funk et al., 2008; Williams and Funk, 2011). Lyon and DeWitt (2012) further examined these persistent droughts and suggested that the decline started abruptly around 1999, a temporally coincident result with the results presented in our study. Other studies linked the drying trend directly to anthropogenic and aerosol forced Indo-PO warming, which results in the westward extension of the Indo-Pacific warm pool causing a westward shift of the Walker circulation and a subsidence anomaly and drying over East Africa (Williams and Funk, 2011; Funk et al., 2014). Hoell et al. (2016) also suggested that human-induced changes in tropical SST exacerbated the effect of natural Pacific decadal variability, enhancing GHA drying during the Long Rain season also.

Despite the numerous studies cited above on teleconnections between the GHA seasonal rains and remote climate anomalies, only a few studies exist on the non-stationarity and decadal changes of teleconnections, and much less in relation to temperatures, the central subject of our analyses. Bahaga et al., 2019 stated that a clear decadal change in the relation between rainfall and the PDO is evident around the early 1920s, the late 1940s, around 1970s, and in recent decades (their Figure 7a), consistent with previously reported changes in the PDO (e.g., Mantua & Hare, 2002). Just last year, also Bahaga et al. 2019 suggested that a potential modulation could occur “driven by the background Pacific decadal variability” and “embedded within the of ENSO forced interannual variability” (Wang et al., 2014; Newman et al., 2016)..

In short, previous research has examined separately particular aspects of the rainfall seasons in Ethiopia and the GHA. Our results are complementary. They concern the overall relationship of regional/local temperatures in the Ethiopian highlands with ENSO dynamics in the Pacific Ocean at the different time scales of relevance. These time scales go from the modulation of seasonality, to the interannual variability, to the decadal (trend-like) changes, embedded within the concurrent variability of the PDO phase. As far as we know, such an integrated picture of regional and global climate has not been addressed for the East African Highlands, and in the specific context of malaria trends. Our findings are consistent with recent studies on the bi-directional interactions between tropical ocean basins, including those between the Pacific and the Western Indian Ocean (WIO). Recent studies highlight how the overall dynamics of the tropical oceans- are tightly linked to global warming (Cai et al., Science 2019)). We have now included a summary in the Discussion to point the reader to literature on links between the

Pacific and the Western Indian Ocean (WIO), including the broader review paper of Cai et al. (Science 2019).

We have added references and text on the literature linking ENSO to the climate of Ethiopia and the Great of Africa. The link to the slowdown is now made in the text (both in the Introduction and the Discussion) through the recognized effect of the El Niño (especially of 1997/98) and the subsequent LN 99/00 on this phenomenon. This link is also made through our findings with the climate simulations on the changing effects of ENSO and the PDO respectively on sea and air temperatures in the region before and after 2000. (Changes that are consistent with the shorter time series analyses). We have also added in the Supplementary Material new results showing the non-stationary association of El Niño (Niño3.4) with SST anomalies in the WIO, as well as with malaria cases. These findings further support the importance of the 97/98 event and the changes post-2000.

As the authors are aware, the PDO and ENSO are not independent of each other. The interannual component of the PDO encapsulates ENSO. The PDO that is said to be linked to the global warming slowdown refers to its decadal component. The PDO signal shown in Figure 4 appears to be the interannual component which should certainly coincide with ENSO. From this, it is not clear how it links the global warming slowdown which is supposed to correspond to the decadal component of the PDO (e.g., see Figure 29a in Santoso et al. 2017 <https://agupubs.onlinelibrary.wiley.com/doi/full/10.1002/2017RG000560>).

We thank the reviewer for this comment which indicates the need for us to further explain this important aspect. Regarding the independence of the PDO and ENSO modes and the extent to which one encapsulates the other, it is well-established that their relation is highly nonlinear, and far from complete. Indeed, the linear correlation between the raw PDO and Niño3.4 time series shown in Fig. 4 for this interval, is only 0.388 and clearly statistically non-significant ($p>0.05$). Even for the same interval, the correlation calculated for the same interannual (IA) frequencies of both ENSO and the PDO is only 0.361, also statistically non-significant (see below the plot with these interannual components for the interval of time used in this study). It is therefore sensible for us to show the PDO IA component together with those of P_f and P_v in Fig 4B, in the sense that the association of this component with the malaria one is not trivially redundant with that of ENSO.

Some studies, in fact, already highlighted the possibility that the late-twentieth-century ENSO activity is anomalously high for at least the past few decades, thus indicating the potential influence of anthropogenic greenhouse warming. (Li et al., 2013; McGregor, Timmermann, et al., 2013). Wang et al. also demonstrated that the frequency of extreme El Niños, not its baseline trend, increases linearly with the GMST and would achieve a doubling for 1.5 °C of warming. But we also concur with the referee that even within the climatological community, these are controversial topics. We have now indicated in our Discussion some open areas of climate research that relate to our findings.

Then in Supplementary Fig. 8, 9 (correlation between ENSO or PDO with grid-point surface air temperature) the authors showed that the influence of ENSO (PDO) weakens (strengthens) after 2000. But taking such a short period (2000-2016) to infer correlation with the decadal component of the PDO is not valid (it is only half the period), and the increased correlation may just be picking up the unresolved near-term trend. Are these correlations statistically significant? Significance test needs to take into account the auto-correlation of the time series which affects the effective degree of freedom (Ebizusaki 1997 <https://journals.ametsoc.org/jcli/article/10/9/2147/28747/A-Method-to-Estimate-the-Statistical-Significance>). Have the seasonal means and trends been removed prior to the correlation? Even the ENSO index contains trends which can affect the correlation in each of the periods. After doing all these, Supp. Fig. 9 and 8 would look similar.

While we agree that intervals are short for climate purposes, and this motivates the use of the atmospheric model in our work, we must clarify that the long-term trends had previously been removed from observations, so that model simulations were implemented without any long-term trends. This methodological aspect is described in the manuscript and in the details of the supplementary section. We explained in our methods that given the short time intervals, model simulations were preferred over working with observations only for the question of changes in PDO and ENSO and their relationship to regional climate in Ethiopia. Therefore, trends should not interfere in any way with the results presented. The main spatial correlation patterns are coherent with the previous results obtained through the analysis of observations, and they are significant, as better explained now in the text. Significant tests that adequately track the decrease of degrees of freedom in line with the randomization test approach that Ebizusaki, 1997 presented, were here used to retrieve significances (values and explanation now included in the Figure captions and in the main text). Further methodological explanation has now been incorporated in the figure captions. We have also edited the text to recognize explicitly the limitation in data length.

How about performing correlation between ENSO index with the local temperature time series before and after 2000, and between the local temperature time series (and ENSO index) vs Pv, Pf?

Thank you for this suggestion. We included instead for the reviewer's reference a table with the relevant correlations among the time series presented in the different figures (the reconstructed components), for the entire intervals of data used in this study. Computing correlations between these components does not require splitting the data at 2000, avoiding the problem of the limited length of the resulting time series. (Part of this table is now in Suppl. Table I).

Variables	Correlation coef.	p-value	Lag	Figure in manuscript
T _{min} , P _v (LT)	0.986	<0.01	0	3A
T _{min} , P _f (LT)	0.978	<0.01	0	3A
P _f , P _v (LT)	0.981	<0.01	0	3A
T _{min} , R (ann)	0.724	<0.01	1	3B
P _f , P _v (ann)	0.956	<0.01	0	3C
T _{min} , P _v (IA)	0.521	<0.05	0	4A
T _{min} , P _f (IA)	0.871	<0.01	0	4A
PDO,N3.4(raw)	0.388	NS	0	4C
PDO,N3.4(IA)	0.361	NS	0	Not shown
HadCRTU4, PDOa	0.051	NS	0	S11B
HadCRTU4, N3.4a	0.384	<0.05	0	S11C

What is rather puzzling is how to connect the trend in local temperature with the recent negative PDO (surface cooling trend over tropical Pacific) when the Indian Ocean has been warming up persistently (e.g., Roxy et al. 2014 <https://journals.ametsoc.org/jcli/article/27/22/8501/34834/The-Curious-Case-of-Indian-Ocean-Warming>). Figure 2d of England et al. (2014; already cited in the text) clearly shows warming over Africa, but cooling of the tropical Pacific. Please clarify and reconcile.

Thank you for this very interesting comment which prompted us to add text in the Discussion on this paper. Although the comment goes beyond the scope of the present study and even into current directions of climate research, we have now referred to Roxy et al. (2014) and attempted to the degree possible in a manuscript on malaria and climate, to provide a sense for the underlying complexity of the climate context. Our central aim was to show how the incidence of two very different malaria parasites in the same area of East Africa coevolve with climate in their

variability and trends at different timescales, first with local temperature and then with the variability of remote large-scale climatic drivers. Ultimately, these drivers are modulated by, and modulate themselves, global warming through a complex set of interhemispheric mechanisms, that remain puzzling even for the most advanced climate scientists. Our study stops in 2007, at a time when human interventions on malaria in the region already mask the synergistic effects of climate variability and change. Unravelling the locally-relevant climatic effects of the Indian Ocean in this complex interplay between the PDO and ENSO and in the context of climate change, is definitely an interesting direction, but beyond the scope of the present study. We added a paragraph in the Discussion with the suggested reference, to highlight the current dynamics indicated by the reviewer and the challenging open questions they raise.

Interestingly, Roxy et al. (2014) provide compelling evidence for a direct contribution from greenhouse warming in the region of interest. They also state that the long-term warming trend over the western Indian Ocean (during summer) is highly dependent on the asymmetry in the El Niño–Southern Oscillation teleconnection (between effects of EN and LN). Note that this is the same area of the Indian Ocean which exhibits high correlations with malaria (also present for air temperatures in T_{2m} —see new Figure S8 in the supplement). These results are in line with our study (Figs. 5 and S8), where the warming of the western tropical IO is clearly seen to be associated with the ENSO forcing. In addition, it has been suggested that a warm Indian Ocean has the potential to weaken the El Niño during its developing and terminating phases (Annamalai et al. 2005; Kug and Kang 2006; Luo et al. 2012), enhancing the probability of a following La Niña episode.

These EN/LN successions correspond exactly to the events we identified as most relevant for malaria dynamics (EN1997/98 and LN99/00, and EN86/87 and LN 88/89), both in the SDC maps for SST (Fig. 5) and T_{2m} (Suppl. Fig. 8) and in the new Supplementary Figure S9. Please see the added results in the correlation maps, and the added Discussion on the climate literature.

The argument that El Nino effect is decreasing after 2000 worth more attention. Apart from weaker El Nino events (e.g., more Central Pacific El Nino as Referee 4 mentioned), the period post 2000 has had more La Nina than El Nino (e.g., Figure 3 in Santoso et al 2017), and that could contribute to more local warming and thus less outbreaks. Reviewer 4 provided references on the change of ENSO which should be considered.

Another interesting comment. We try now to be more precise as we did not intend to state that El Niño effects were decreasing over the region up to the present. We only interpreted the results of both the analysis and simulations with reference to the interval of time covered, up to 2007 and not beyond this year. In this regard, the referee will know that climate in the last decade has exhibited anomalies not yet fully understood in different ocean basins of relevance to our study. We agree that there have been a series of recent LN events, but this fact would not detract from the main findings in our study. In fact, Roxy et al. 2014 also state that the effects of LN are not opposite to those of EN (the asymmetry we mentioned above). Unfortunately, though, the widespread intervention campaigns that started in 2005 make it now more difficult to tease apart these effects of climate variability on malaria dynamics. This reality further underscores the need to keep monitoring the dual roles of climate and malaria control in this highland region, especially given the possibility of relaxing control efforts.

As we wished however to address as much as possible the concerns of the reviewer, we expand here on aspects of changes in ENSO and their connections to global climate, including

interactions with the Pacific and Indian Ocean. On some of this basis, we have written the much shorter discussion in the edited text.

In a series of previous studies, we addressed the complex interactions between Pacific Ocean SST and the dynamics of the Indian Ocean (IO) basin. In Cash et al. 2008 and Cash et al., 2009, we showed by means of an ensemble pacemaker experiment that by prescribing surface conditions in different ocean regions in the Pacific Ocean, it was possible to generate coherent atmospheric and oceanic signatures similar to those seen for El Niño and La Niña episodes. A similar atmospheric model than the one used here was applied in that work. These responses to EN and LN episodes were seen in both sea and air surface temperatures over the western IO (WIO). The composite pattern below for EN-LN events with warming over the WIO, clearly resembles the structure found for the SDC map of correlations between P_f malaria and SST in Fig. 5 (and also for the new results for P_f and T_{2m} in the new Supp. Fig. 8). This correspondence in spatial structure is an indication that similar processes in the climate are involved with the forcing of malaria dynamics.

In terms of mechanism, SST anomalies induce an increase in geopotential heights in the tropics during the NH summer months following a winter El Niño. The pattern expands through the spring representing the wintertime height response to the change in SST. During DJF, increases in lower-tropospheric heights are apparent all across the tropics (not shown, but we refer the referee to the above references), including the region in East Africa covered by our study. These dynamics are consistent with a shift in the Walker circulation during El Niño years, a result also seen with previous models and observations (Alexander et al. 2002). In fact, the warming of the western tropical Indian Ocean (WTPO) mentioned earlier by the referee, has been shown to be weak and much less influential over the WTIO region than the eastern and central Pacific SST anomalies (Fig. above in Cash et al. 2009). These results are fully aligned with what we showed for the combined effects over Ethiopia of the EN events under the decadal modulation by the PDO.

Interestingly, Roxy et al. (2014) in their study of SST trends for the past century, showed a long-term warming trend over the western Indian Ocean that is larger than that over the warm pool in both magnitude and period (their Fig. 1c). In their Fig. 6a, these authors compute the difference in the SST over the Indian Ocean for the NH summer, between the periods of 1951–2012 and 1901–50, obtaining a pattern that strongly resembles the one we present above and in the composite average in Supp. Fig. 9. Moreover, the results from that study emphasize the

asymmetry in the ENSO teleconnection as one of the reasons why El Niño events induce anomalous warming over the western Indian Ocean (as shown both in our study and in Fig. 1C of Roxy et al., 2014). As they also show, La Niña events fail to do the inverse. With data analyses up to 2012, the authors state that the frequency of El Niño events has increased during recent decades, and highlight some important conclusions by the Intergovernmental Panel on Climate Change Fifth Assessment Report (IPCC AR5) that are of relevance for our reply (and we have included in our Discussion). Namely, the IPCC AR5 points out that 90% of the heat resulting from global warming during the last four decades has been accumulated in the oceans (Rhein et al. 2014), and that the periodic occurrence of El Niño acts as a vent to exchange this heat from the ocean to the atmosphere. It is this heat that is partially transferred to the Indian Ocean via a modified Walker circulation, and is reflected in the warming trend over the region and in the alteration of the overall Pacific basin dynamics. It is interesting to note as the IPCC remarks do, that the warming trend over the Indian Ocean is a major contributor to, and largely in phase with, the overall trend in the global mean SST (Rhein et al., 2014 and Fig. 7 in Roxy et al., 2014).

In addition, the referee would agree that several references (including the ones in the above paragraphs) show that irrespective of LN dynamics, the frequency of El Niño events has increased in the recent decades, but that a strong warm event has not been recorded since 1997/98. This absence of a major event may be the reason why the Pacific and Indian Ocean SST anomalies show a slight dampening, possibly contributing to the recent hiatus in the global surface warming (Kosaka and Xie 2013). In short, the recent cool conditions over the eastern Pacific might be due to the feedback from a warmer Indian Ocean, bringing the sequence of cause and effects into a vicious cycle, which requires further research. Several other studies have also noted that the warming trends over the Indian and Atlantic Oceans lead to La Niña-like conditions over the Pacific (Kucharski et al. 2011; Kug and Kang 2006; Luo et al. 2012).

We therefore agree that changes in ENSO involve complex global linkages, more so in relation to global warming, which we did not intend to minimize. In particular, we see no contradiction between our findings on connections between ENSO and air/sea temperatures in the region of interest in East Africa and the Western Indian Ocean, with the existing literature on global warming and its connections to EN/LN events. On the contrary, we think our findings are consistent and reinforced by this context. But this is clearly an area that climate research should continue to elucidate, including the complex interactions that can generate positive feedbacks and may appear contradictory. We hope the added text and references in the Discussion suffice to make this clearer and to direct the interested reader to relevant climate literature.

In Summary, the present manuscript has not clearly linked the downward trend in malaria case to the global warming slowdown.

We thank the referee for this frank criticism even though we were somewhat surprised by it. We interpreted it as resulting primarily from our writing and from not providing sufficient context. We hope the expanded text and results do address this concern together with the clarifications below. We elaborate below on what we do in the work, and on more conceptual aspects of this question. We also offer the possibility of changing the title if the current one could be mis-interpreted in terms of the content of our work to:

“Turn-of-the-century slowdown in highland malaria trend reflects coupling to climate variability and change”

The referee seems to refer in this comment to the part of the paper linking the regional to the global climate, as the evidence presented for an effect of temperatures on malaria locally in

these Ethiopian highlands is not into question. That is, we interpret the comment by the referee as specifically referring here to the part of the work which then expands into global couplings to the Pacific Ocean. In this regard, we would like to briefly summarize here the components of our manuscript. After (1) the process-based model of malaria transmission driven by local temperatures and (2) the statistical analyses of associations between local variables at different temporal scales, we transition specifically to the role of ENSO (and the PDO) on the regional climate and on malaria. The motivation for this focus is explained above and in the paper (the key role of these Pacific Ocean phenomena in the global climate, including their documented links to the slowdown in global warming). We complement statistical analyses with the simulations of the global climate model (in particular to be able to consider a longer time period when addressing the links between interannual climate variability in the Pacific and the regional climate of East Africa including the adjacent part of the Indian Ocean). These simulations indicate that at the turn of the century the influence of ENSO weakened (whereas that of the PDO strengthened). In this sense, we claim to address the slowdown, regionally and globally.

We agree that the role of temperature and climate drivers we have presented does not consider directly correlations with a global time series of temperatures. But we do not think this is what the referee would be proposing. Comparison of trends as an approach to link global warming to a region's climate conditions is known to be problematic. Calculating correlations between trends is possibly not the best way to assess that two processes are interdependent or that one drives the other. Moreover, the distinction between the increase in GMST and global warming (GW) itself, as a suite of processes and associated phenomena, becomes important. The latter includes the former as a manifestation, but the two are not equivalent. As the referee knows well, GW, is indeed a much broader concept whose effects are not just seen as increases in the average temperature of the planet (GMST as a measure of global trend), but in the many effects at different temporal and spatial scales on other variables (precipitation, humidity, sea-level rise, winds and large-scale regional atmospheric circulation, etc...). No one would expect that all regions in the world manifest effects of global warming via exactly the same trends in temperature or in other climate variables. There is for instance, no 'precipitation warming' congruent with GW; in some regions, precipitation decreases whereas in others it increases. Similarly, there is no unique pattern of 'temperature warming' that applies equally everywhere. There is instead a global warming phenomenon that affects the planet, evident globally in an increase in GMST, but which affects in different ways the frequency and intensity of climate processes operating at a wide range of spatial and temporal scales. As a result, there is a nonlinear array of changes in temperature at smaller scales than the globe (regional, local) of variable steepness (and even including local cooling). Some places warm up much more than the global average and others much less, and this difference is not due to inconsistencies in the relationship with GW or a lack of it. Similarly, precipitation trends, both increases and decreases, are all linked to GW, and the patterns are also well-known and correctly simulated by climate change projections. For instance, GW is known to affect climate variability patterns and the frequency of extreme events and of climate modes (ENSO, NAO, IDM, PDO, etc...), including in ways that we do not yet necessarily understand well, even when the influence itself is unquestioned. To complicate matters further, GW has direct and indirect effects, via delayed responses that will manifest themselves in the future. Therefore, even if global warming were halted today (e.g. no more increases in the global average temperature trend), the risk associated with increased frequency of extreme El Niño events may still continue for several decades (Power et al., 2017; Wang et al., 2017). For the above reasons, the steps we followed tried to address the relevant effect of GW in parts of the African continent with a more

integrated approach. As a result of all of these complexities, simple correlations between the GMST trend and a local temperature trend can be simplistic and even misleading.

We know the referee is fully aware of these complexities and issues, and did not ask for the direct consideration of a global temperature time series. Our intent here is to emphasize why we adopted what we considered a more integrated approach to the effect of the slowdown in warming on highland malaria in Ethiopia, with the steps we summarized above. What we recognize was missing was to better present the context for the focus on the Pacific Ocean. Finally, we would like to close this response with why we think the findings are important in terms of public health, and provide an improved understanding of highland malaria in the context of climate conditions: (1) there was a long-lasting debate in the 1990s and 2000s on trends in temperatures and highland malaria that is of relevance to the large populations of these regions. This controversy has taken many forms and was not completely resolved, to the detriment of the proper consideration of how changes in climate interact with control efforts. An important open question is what happened at the turn of the century. (2) Despite the knowledge on climate controls known to operate over the region and on how large-scale climate regimes influence the vast droughts that impact the area and cause extensive famines with massive health effects, the two worlds of climate and epidemiology have not sufficiently come together. In this context, we believe that despite the considerable efforts of a number of public health scientists, the importance of integrating climate conditions into the malaria intervention portfolio is not sufficiently appreciated by malaria public health agencies/local governments/international agencies-donors. (3) The interplay of climate and malaria dynamics is particularly relevant in light of possible relaxation of control strategies following widespread intervention campaigns. We have documented pernicious effects of this kind of cycle in the context of interannual variability (for rainfall) in another unstable malaria region, in semi-deserts of India. Consideration of long-term, decadal, variation of the climate has not been raised, as far as we know, despite what we believe to be important implications for the sustained success of control efforts. We hope this short summary helps place the work in this perspective.

Other comments:

L33: “output of numerical simulations of a global climate model” – note that CAM is an atmospheric model, not a full climate model. An atmospheric model has no full interactions with the ocean, and the sea surface temperatures are prescribed rather than freely evolving. The authors ‘examined the consensus of atmospheric model simulations’ (L236), how many ensemble members are examined?

Thank you. We have now corrected this first sentence and also specified that we analyzed the consensus simulation of a 10-member ensemble.

Which temperature variable goes into the dynamical disease model? The one in Fig. 1B or the reconstructed components?

The yellow one in Fig. 1B. Definitely not the reconstructed components. See model description. We used the reconstructed components to specifically address statistical associations of variability at similar scales. These explorations of pattern motivated the modeling work for which we introduce the temperature without any filtering.

L186-190: Need to elaborate here what ENSO and PDO actually do to climate in the region of study.

Thank you. This is a suggestion that we much appreciated as expressed in many of the responses to the questions above. In fact, we realized we had not elaborated on this in our former version of the manuscript. We now have added text.

L89 would read better like this: “In particular, a process-based transmission model is used to predict what would have been the effect of the observed change in temperatures on falciparum malaria cases in the absence of the public health interventions introduced post-2004.”

Ok thank you for the suggestion. Changed accordingly.

L69-76: “The warming slowdown.... in addition to a result of statistical discrepancies from different data sets and definitions of the phenomenon... Irrespective of the final causes behind it, the documented global took place..” - this part needs a reword. Statistical discrepancies and definitions cannot physically cause the warming slowdown that actually took place.

Ok, re-written accordingly, thank you.

L121: “real trend” – not clear

Thank you; we replaced the term ‘real trend’ by ‘extracted trend’.

L232 and elsewhere: “HadCRTU4” -> “HadCRUT4”

Agreed, thank you for highlighting the typo!

Supp. Figure 8-9: “Spatial correlations” – the analysis does not show a spatial correlation. It shows temporal correlation between the index and surface air temperature at each grid-point on the map.

We rewrote the sentence to more accurately reflect what was done.

L249: “Decadal variability in malaria cases for both parasites, P. falciparum and P. vivax, tracks the trends”. – Not apparent in Figure 1

As stated before, we have now re-drawn Figure 1 to better support this sentence.

Scales in the different variables to include them all in the same graph do not help.

We tried to understand what was meant here. Unfortunately, we cannot follow enough to respond. We do not know which graph this refers to, as figures 2 (now figure 3) and 4 have plots that correspond to one characteristic time scale at a time. It is true that we consider different variables, namely malaria cases, temperatures, ENSO and PDO indices, but all local/regional variables are now in Figure 3 (previously 2) and the global climate ones were moved to Fig 4.

References:

Bahaga, T., Fink, A., Knippertz, P. (2019). Revisiting interannual to decadal teleconnections influencing seasonal rainfall in the Greater Horn of Africa during the 20th century. *Int. J. Climatol.* 39, 5: 2765-2785.

Beltrando, G. and Camberlin, P. (1993), Interannual variability of rainfall in the eastern horn of Africa and indicators of atmospheric circulation. *Int. J. Climatol.*, 13: 533-546.

Broomhead D.S., King G.P. Extracting qualitative dynamics from experimental data. *Physica D.* 1986;20:217–236.

Camberlin, P., and N. Philippon, 2002: The East African March–May Rainy Season: Associated Atmospheric Dynamics and Predictability over the 1968–97 Period. *J. Climate*, 15, 1002–1019, [https://doi.org/10.1175/1520-0442\(2002\)015<1002:TEAMMR>2.0.CO;2](https://doi.org/10.1175/1520-0442(2002)015<1002:TEAMMR>2.0.CO;2).

Carroll et al., 2016. Scale-dependent complementarity of climatic velocity and environmental diversity for identifying priority areas for conservation under climate change. *Glob Change Biol.* 2017;23:4508–4520.

Cash, B., Rodó, X., Kinter, J. (2008). Links Between Tropical Pacific SST and Cholera Incidence in Bangladesh: Role of the Eastern and Central Tropical Pacific. *J. Clim.* 21: 4647-4663.

Cash, B., Rodó, X., Kinter, J. (2009). Links Between Tropical Pacific SST and Cholera Incidence in Bangladesh: Role of the Western and Central Extratropical Pacific. *J. Clim.* 22: 1641-1660.

Cazelles B, Chavez M, McMichael AJ, Hales S (2005) Nonstationary Influence of El Niño on the Synchronous Dengue Epidemics in Thailand. *PLoS Med* 2(4): e106. <https://doi.org/10.1371/journal.pmed.0020106>

Cazelles B, Chavez M, Magny GC, Guégan JF, Hales S. Time-dependent spectral analysis of epidemiological time-series with wavelets. *Journal of the Royal Society, Interface.* 2007 Aug;4(15):625-636. DOI: 10.1098/rsif.2007.0212.

Diatla, S. and Fink, A.H. (2014), Statistical relationship between remote climate indices and West African monsoon variability. *Int. J. Climatol.*, 34: 3348-3367.

Diro, G.T., Black, E. and Grimes, D.I.F. (2008), Seasonal forecasting of Ethiopian spring rains. *Met. Apps*, 15: 73-83. doi:10.1002/met.63

Diro, G.T. DiroD., I.F., Grimes, E., Black, Emily (2010). Teleconnections between Ethiopian summer rainfall and sea surface temperature: Part I-observation and modelling. *Clim. Dyn.* 37(1):103-119.

Du L.P., Wu S.H., Liew A.W.C., Smith D.K., Yan H. Spectral analysis of microarray gene expression time series data of *Plasmodium falciparum*. *IJBRA.* 2008;4(3):337–349.

Funk, C., Dettinger, M.D., Michaelsen, J., Verdin, J. et al. (2008). Warming of the Indian Ocean threatens eastern and southern African food security but could be mitigated by agricultural development. *Proc. Nat. Acad. Sci.* 105 (32) 11081-11086.

- Funk, C., Hoell, A., Shukla, S., Bladé, I., Liebmann, B., Roberts, J.B., Robertson, F.R., Husak, G. (2014). Predicting East African spring droughts using Pacific and Indian Ocean sea surface temperature indices. *Hydrol. Earth Syst. Sci.*, 18, 4965–4978.
- Gissila, T., Black, E., Grimes, E., J. M. Slingo (2004). Seasonal forecasting of the Ethiopian Summer rains. *Int. J. Clim.* 24(11) DOI: 10.1002/joc.1078.
- Golyandina N., Nekrutkin V., Zhigljavsky A.A. (2010) Analysis of time series structure: SSA and related techniques. CRC Press.
- Hansen, J. Sato, M., Kharecha, P., von Schuckmann, K. (2011). Earth's energy imbalance and implications. *Atmos. Chem. Phys.*, 11, 13421–13449.
- Hassani H. (2007). Singular spectrum analysis: methodology and comparison. *JDS*, 5:239–257.
- Hassani, H, Ghodsi, Z. (2015). A glance at the applications of Singular Spectrum Analysis in gene expression data. *Biomol Detect Quantif.* Jun; 4: 17–21.
- Hastenrath, S., D. Polzin, and C. Mutai, 2011: Circulation Mechanisms of Kenya Rainfall Anomalies. *J. Climate*, 24, 404–412.
- Hoell, A., Hoerling, M., Eischeid, Quan, X., Liebmann, B. (2016). Reconciling Theories for Human and Natural Attribution of Recent East Africa Drying. *J. Clim.* 30(6) DOI: 10.1175/JCLI-D-16-0558.1.
- Jury, M. (2010). Ethiopian decadal climate variability. *Theor. Appl. Clim.* 101(1):29-40.
- Korecha, D. and Barnston, A.G. (2007) Predictability of June-September Rainfall in Ethiopia. *Monthly Weather Review*, 135, 628-650.
- Korecha, D., and Sorteberg, A. (2013), Validation of operational seasonal rainfall forecast in Ethiopia, *Water Resour. Res.*, 49, 7681– 7697, doi:10.1002/2013WR013760.
- Kosaka, Y., and S.-P. Xie, 2013: Recent global-warming hiatus tied to equatorial Pacific surface cooling. *Nature*, 501, 403–407.
- Kucharski, F., I. S. Kang, R. Farneti, and L. Feudale, 2011: Tropical Pacific response to 20th century Atlantic warming. *Geophys. Res. Lett.*, 38, L03702, doi:10.1029/2010GL046248.
- Kug, J.-S., and I.-S. Kang, 2006: Interactive feedback between ENSO and the Indian Ocean. *J. Climate*, 19, 1784–1801.
- Li, J., Xie, S.-P., Cook, E. R., Morales, M. S., Christie, D. A., Johnson, N. C., ... D'Arrigo, R. (2013). El Niño modulations over the past seven centuries. *Nature Climate Change*, 3(9), 822–826
- Luo, J.-J., W. Sasaki, and Y. Masumoto, 2012: Indian Ocean warming modulates Pacific climate change. *Proc. Natl. Acad. Sci. USA*, 109, 18 701–18 706.
- Lyon, B., and DeWitt, D. G. (2012), A recent and abrupt decline in the East African long rains, *Geophys. Res. Lett.*, 39, L02702, doi:10.1029/2011GL050337.
- Mantua, N., Hare, S. 2002. The Pacific Decadal Oscillation. *Journal of Oceanography*, 58: 35-44.

McGregor, S., Timmermann, A., England, M. H., Timm, O., & Wittenberg, A. T. (2013). Inferred changes in El Niño–Southern Oscillation variance over the past six centuries. *Climate of the Past*, 9(5), 2269–2284

Mutai, C., Ward, Neil. (2000). East African Rainfall and the Tropical Circulation/Convection on Intraseasonal to Interannual Timescales. *J. Clim.*, 13(22):3915-3939.

Nicholson, S. E. (2017), Climate and climatic variability of rainfall over eastern Africa, *Rev. Geophys.*, 55, 590– 635

Ogallo, L.J. (1988), Relationships between seasonal rainfall in East Africa and the Southern Oscillation. *J. Climatol.*, 8: 31-43.

Power, S. P., Delage, F., Chung, C., Ye, H., & Murphy, B. (2017). Humans have already increased the risk of major disruptions to Pacific rainfall. *Nature Communications*, 8.

Remais, J. et al. 2009. Model approaches for estimating the influence of time-varying socio-environmental factors on macroparasite transmission in two endemic regions. *Epidemics* 1, 4: 213-220.

Rhein, M., and Coauthors, 2014: Observations: Ocean. *Climate Change 2013: The Physical Science Basis*, T. F. Stocker et al., Eds., Cambridge University Press, 255–315.

Roxy, M., Ritika, K., Terray, P., Masson, S. (2014). The Curious Case of Indian Ocean Warming. *J. Clim.*, 27:8501-8509.

Rodó, X., Roger Curcoll, Marguerite Robinson, Joan Ballester, Jane C. Burns, Daniel R. Cayan, W. Ian Lipkin, Brent L. Williams, Mara Couto-Rodriguez, Yosikazu Nakamura, Ritei Uehara, Hiroshi Tanimoto, Josep-Anton Morguí (2014). Winds from NE China carry the KD agent to Japan. *Proc. Nat. Acad. Sci.*, 111 (22): 7952-7957.

Ropelewski, C. F., and M. S. Halpert, 1987: Global and Regional Scale Precipitation Patterns Associated with the El Niño/Southern Oscillation. *Mon. Wea. Rev.*, 115, 1606–1626,

Sanei S., Ghodsi M., Hassani H. An adaptive singular spectrum analysis approach to murmur detection from heart sounds. *IPEM*. 2011;33(3):362–367.

Segele ZT, Lamb PJ, Leslie LM (2009a) Large-scale atmospheric circulation and global sea surface temperature associations with Horn of Africa June–September rainfall. *Int. J. Climatol.* 29:1075–1100.

Seleshi Y, Demaree GR (1995) Rainfall variability in the Ethiopian and Eritrean highlands and its links with the southern oscillation index. *J Biogeogr* 22:945–952. doi:10.2307/2845995.

Suárez-Moreno, R., Rodríguez-Fonseca, B., Barroso, J., Fink A. (2018). Interdecadal Changes in the Leading Ocean Forcing of Sahelian Rainfall Interannual Variability: Atmospheric Dynamics and Role of Multidecadal SST Background. *J. Clim.* 31(17).

Tang V.T., Yan H. Noise reduction in microarray gene expression data based on spectral analysis. *IJMLC*. 2012;3(1):51–57.

Vautard R., Yiou P., Ghil M. Singular-spectrum analysis: a toolkit for short, noisy chaotic signal. *Physica D*. 1992;58:95–126.

Vigaud, N., Robertson, A.W., Tippett, M.K. and Acharya, N. (2017) Subseasonal Predictability of Boreal Summer Monsoon Rainfall from Ensemble Forecasts. *Front. Environ. Sci.* 5:67. doi: 10.3389/fenvs.2017.00067

Wang, G., Cai, W., Gan, B., Wu, L., Santoso, A., Lin, X., ... McPhaden, M. (2017). Continued increase of extreme El Nino frequency long after 1.5°C warming stabilization. *Nature Climate Change*, 7(8), 568–572

Williams, A.P., Funk, C. A. (2011). Westward extension of the warm pool leads to a westward extension of the Walker circulation, drying eastern Africa. *Clim Dyn* 37, 2417–2435 .

Xie H.B., Guo T., Sivakumar B., Liew A.W.C., Dokos S. Symplectic geometry spectrum analysis of nonlinear time series. *Proc R Soc A*. 2014;470(2170):20140409.

Yeh, S.-W., Cai, W., Min, S.-K., McPhaden, M. J., Dommenges, D., Dewitte, B., ... Kug, J.-S. (2018). ENSO atmospheric teleconnections and their response to greenhouse gas forcing. *Reviews of Geophysics*, 56, 185– 206. <https://doi.org/10.1002/2017RG000568>

Zhang, R., Gao, C., Kang, X. et al. ENSO Modulations due to Interannual Variability of Freshwater Forcing and Ocean Biology-induced Heating in the Tropical Pacific. *Sci Rep* 5, 18506 (2016). <https://doi.org/10.1038/srep18506>

Reviewers' Comments:

Reviewer #1:

Remarks to the Author:

The latest version of the manuscript has significantly improved. The authors have now addressed my major concerns. I just have a few final minor comments, once they're addressed the paper can be accepted for publication in Nature communication. My sincere congratulations to the authors.

Minor comments:

Abstract:

"and potentially complacent plans for relaxing control" – reword as control efforts will still have to be conducted in the field to limit the spread of malaria – it sounds like "our model shows less risk so fine so no need for control". This statement is not acceptable so please reword

Text:

"in the temporal patterns of incidence" – that paper was based on the theoretical length of the transmission season really – perhaps "malaria risk" instead but fine

L150 – "at interannual time scale" – singular

L165 – "are obtained for a model driven by Tmin"

L213 – during the late 1980s and 1990s.

L241 – warm ENSO conditions.

L404 – "underlines this point further" sounds better

L450 – please precise spatial resolution of NCEP data (as you already do for CRUTS4.01)

L457: Use degree symbol in the text for lat/lon coordinates (5°N-5°S)

Reviewer #6:

Remarks to the Author:

The authors are thanked for their thoughtful revision. There are still a few remaining points below that were not fully addressed.

L205-208: "Moreover, both ENSO and the PDO are known to influence the seasonal migration of the Intertropical Convergence Zone (ITCZ), and ultimately the seasonal-to-interannual variability in rainfall and the variation in temperatures in eastern Africa³⁹." What I meant in my previous comment was for the author to state explicitly the particular effect of El Nino, e.g., what direction does the ITCZ shift during El Nino, and are the specific effects on rainfall and temperature response over the studied region. Similarly, state for the positive (or negative) phase of the PDO. This is for the readers to better understand the climatic effect on Malaria outbreaks.

Please elaborate how the statistical significance in Sup. Fig. 12 and 13 is computed. There are two factors to consider in this case: 1) the number of ensemble members, 2) auto-correlation (which reduces the effective degree of freedom). The latter is especially relevant for the PDO which contains significant decadal variability, meaning that for the PDO, the cut-off correlation coefficient at p-value 0.05 should be larger than that for Nino3.4 (0.226, Sup. Fig. 12). Did the authors consider both factors? Factor #1 should help compute the statistical significance in terms of model agreement. It would help to stipple the locations where the correlations are significant. The authors nonetheless already put a caveat on this, e.g., L356-358, but still a clear evaluation is warranted as the PDO contains low-frequency variability such that a correlation over a relatively short period 1979-2000 and 2001-2016 would appear high even though not statistically significant.

Supplementary Fig. 13 caption: "spatial correlations" - Please reword. This is not spatial correlation, but temporal correlation at each spatial location.

L77: "El Niño on record for 1998 followed also by a strong La Niña" should be reworded to: "El Niño on record in 1997-1998 followed by two consecutive strong La Niña events". Note that 1998-1999 and 1999-2000 La Nina events were both strong. A reference is needed here which can also be used as a reference at other instances, e.g., L217 on the amplitude of El Nino events of 1997-1998, 2002-2003, and 2006-2007. It should be clarified that the 2002-2003 and 2006-2007 events are not "large" El Nino as stated in L217 (see Fig. 3 of Santoso et al. 2017, <https://agupubs.onlinelibrary.wiley.com/doi/full/10.1002/2017RG000560>). They are in fact weak El Ninos, but it is OK to reword "large" to "significant" (as in they are clearly distinguishable from neutral, i.e., as opposed to events indicated by empty circles in their Fig. 3).

The manuscript title: "the 2000 'slowdown' in global warming" is not clear.
It might be better: "Malaria trends in Ethiopian highlands track the early 21st Century global warming slowdown"

Supp. Fig. 8: missing x-axis labels in the panels.

Reviewer #1 (Remarks to the Author):

The latest version of the manuscript has significantly improved. The authors have now addressed my major concerns. I just have a few final minor comments, once they're addressed the paper can be accepted for publication in Nature communication. My sincere congratulations to the authors.

We are glad to hear, and we thank the referee for his/her role in improving the manuscript.

Minor comments:

Abstract:

“and potentially complacent plans for relaxing control” – reword as control efforts will still have to be conducted in the field to limit the spread of malaria – it sounds like “our model shows less risk so fine so no need for control”. This statement is not acceptable so please reword

We did not mean that our model shows less risk but that the risk can be mis-interpreted of one does not take into consideration the climate conditions, and that the resulting over-estimation of the effects of control, may lead to misguided policies to relax intervention efforts. We found no way to include the longer text the referee suggests, and it is not exactly what we are trying to say. We did now reworded this sentence and refer to “potentially misguided or premature plans for relaxing control”. Hopefully, this change now better conveys the meaning.

Text:

“in the temporal patterns of incidence” – that paper was based on the theoretical length of the transmission season really – perhaps “malaria risk” instead but fine
We now used malaria risk as suggested.

L150 – “at interannual time scale” – singular
Ok.

L165 – “are obtained for a model driven by Tmin”
Right.

L213 – during the late 1980s and 1990s.
Ok.

L241 – warm ENSO conditions.
Done.

L404 – “underlines this point further” sounds better
Done.

L450 – please precise spatial resolution of NCEP data (as you already do for CRUTS4.01)
Thank you for asking: 2.5 deg. resolution, as now indicated.

L457: Use degree symbol in the text for lat/lon coordinates (5°N-5°S)
Done.
Thank you.

Reviewer #6 (Remarks to the Author):

The authors are thanked for their thoughtful revision. There are still a few remaining points below that were not fully addressed.

We thank the referee for acknowledging the work made to accommodate his/her former requests.

L205-208: *“Moreover, both ENSO and the PDO are known to influence the seasonal migration of the Intertropical Convergence Zone (ITCZ), and ultimately the seasonal-to-interannual variability in rainfall and the variation in temperatures in eastern Africa³⁹.”* What I meant in my previous comment was for the author to state explicitly the particular effect of El Niño, e.g., what direction does the ITCZ shift during El Niño, and are the specific effects on rainfall and temperature response over the studied region. Similarly, state for the positive (or negative) phase of the PDO. This is for the readers to better understand the climatic effect on Malaria outbreaks.

We thank the referee for the further clarification of this point. We have now added text both here in the Results and in the Discussion to provide further information.

Please elaborate how the statistical significance in Sup. Fig. 12 and 13 is computed. There are two factors to consider in this case: 1) the number of ensemble members, 2) auto-correlation (which reduces the effective degree of freedom). The latter is especially relevant for the PDO which contains significant decadal variability, meaning that for the PDO, the cut-off correlation coefficient at p -value 0.05 should be larger than that for Niño3.4 (0.226, Sup. Fig. 12). Did the authors consider both factors? Factor #1 should help compute the statistical significance in terms of model agreement. It would help to stipple the locations where the correlations are significant. The authors nonetheless already put a caveat on this, e.g., L356-358, but still a clear evaluation is warranted as the PDO contains low-frequency variability such that a correlation over a relatively short period 1979-2000 and 2001-2016 would appear high even though not statistically significant.

As explained, correlations were computed between the individual detrended time series of Niño3.4 (or the PDO) and the ensemble average at each point in the grid map. To clarify we added some brief text on this point in the caption of Suppl. Fig. 12 and directed the reader in the caption of Suppl. Fig. 13 to this other caption.

Following an earlier request of the referee, we had already included in the previous revision a new table (Suppl. Table I), with the correlation values between detrended components in both malaria and climate. We had also described how the significance levels were obtained with a randomization test that accounted for the decrease in the d.o.f. Results indicated very strong associations.

It must be noted that the PDO component we used for the computation of correlation maps in Suppl. Fig. 13 corresponds to the detrended time series shown in Suppl. Fig. 11, and the correlations have already taken into account the decrease in the d.o.f. in the calculations of the significance thresholds provided for Suppl. Fig. 12 and 13. As the referee can see in Suppl. Fig. 11, this component for the PDO displays roughly the same variability ranges as those for EN. The referee is nevertheless right in indicating that we had not specified the specific threshold value for the significance with the PDO in Suppl. Fig. 13. This value has now been

added in the caption. However, as the referee will also see, correlations over Ethiopia are above this value for the pre-2000 simulations, and even further above this threshold for the post-2000 simulations (above 0.6 nearly in all country and over 0.7 in much of the affected Oromia region).

We would like to clarify that the significance is determined in two steps: first, on a point-wise basis for each individual grid point in the map (accounting for the mentioned decrease in d.o.f.), and second, by assessing the non-randomness of the spatial pattern by means of a randomization test. We added a sentence to clarify this point in the captions corresponding to both figures.

Supplementary Fig. 13 caption: "spatial correlations" - Please reword. This is not spatial correlation, but temporal correlation at each spatial location.

Agreed, we changed this sentence accordingly

L77: "El Niño on record for 1998 followed also by a strong La Niña" should be reworded to: "El Niño on record in 1997-1998 followed by two consecutive strong La Niña events". Note that 1998-1999 and 1999-2000 La Nina events were both strong.

Thank you, we changed this text accordingly.

A reference is needed here which can also be used as a reference at other instances, e.g., L217 on the amplitude of El Nino events of 1997-1998, 2002-2003, and 2006-2007. It should be clarified that the 2002-2003 and 2006-2007 events are not "large" El Nino as stated in L217 (see Fig. 3 of Santoso et al. 2017, <https://agupubs.onlinelibrary.wiley.com/doi/full/10.1002/2017RG000560>). They are in fact weak El Ninos, but it is OK to reword "large" to "significant" (as in they are clearly distinguishable from neutral, i.e., as opposed to events indicated by empty circles in their Fig. 3).

We reworded the sentence accordingly and added the suggested reference, thank you.

The manuscript title: "the 2000 'slowdown' in global warming" is not clear.

It might be better: "Malaria trends in Ethiopian highlands track the early 21st Century global warming slowdown"

Thank you for the suggested title, which we have now adopted.

Supp. Fig. 8: missing x-axis labels in the panels.

Done.

Reviewers' Comments:

Reviewer #6:

Remarks to the Author:

The authors have addressed previous comments. I am pleased to recommend acceptance.